# Pretraining Codomain Attention Neural Operators for Solving Multiphysics PDEs

**Md Ashiqur Rahman**[1], **Robert Joseph George**[2], **Mogab Elleithy**[2], **Daniel Leibovici**[2],
**Zongyi Li**[2], **Boris Bonev**[3], **Colin White**[2], **Julius Berner**[2], **Raymond A. Yeh**[1],
**Jean Kossaifi**[3], **Kamyar Azizzadenesheli**[3], **Anima Anandkumar**[2]
[1]Purdue University, [2]Caltech, [3]NVIDIA

## Abstract

Existing neural operator architectures face challenges when solving multiphysics problems with coupled partial differential equations (PDEs) due to complex geometries, interactions between physical variables, and the limited amounts of high-resolution training data. To address these issues, we propose *Codomain Attention Neural Operator* (CoDA-NO), which tokenizes functions along the codomain or channel space, enabling self-supervised learning or pretraining of multiple PDE systems. Specifically, we extend positional encoding, self-attention, and normalization layers to function spaces. CoDA-NO can learn representations of different PDE systems with a single model. We evaluate CoDA-NO's potential as a backbone for learning multiphysics PDEs over multiple systems by considering few-shot learning settings. On complex downstream tasks with limited data, such as fluid flow simulations, fluid-structure interactions, and Rayleigh-Bénard convection, we found CoDA-NO to outperform existing methods by over $36\%$.

## 1 Introduction

Many science and engineering challenges involve solving partial differential equations (PDEs). A PDE can represent physical phenomena such as fluid dynamics, wave propagation, material deformation, etc., but to describe many real-world systems, multiple such PDEs must be coupled together, viz., multi-physics modeling [1]. For instance, in subsurface engineering, equations of flow, thermodynamics, and microchemistry are coupled together [2]; in materials science, physics at multiple scales are involved in modeling [3], and in weather forecasting, atmospheric processes involve interactions of wave propagation and fluid dynamics [4].

Traditionally, numerical methods have been devised to solve PDEs. However, they typically require discretization of PDEs on fine grids to capture the physical phenomena accurately. Consequently, these computational requirements often exceed available memory and computational budgets for real-world applications. Beyond these obstacles present in individual PDE problems, the convergence of numerical solvers in multiphysics systems presents major difficulties arising from intricate interactions among multiple coupled PDEs.

Deep learning techniques have emerged as faster alternatives to numerical solvers for PDEs in many applications. They are typically trained using supervised learning with data obtained from solvers. This becomes a challenge when only limited data is available, especially in the case of multiphysics simulations, which are expensive and challenging for numerical solvers. Instead, obtaining data from simpler simulations where only a subset of the "physics" is incorporated is more convenient and less expensive. In other words, instead of getting data from coupled PDE systems, we can obtain data by solving individual PDEs. While the solutions of the two systems can be very different, they share common features and can benefit from a combined learning framework. *Can we design a systematic curriculum learning scheme for learning multiphysics systems?*

38th Conference on Neural Information Processing Systems (NeurIPS 2024).

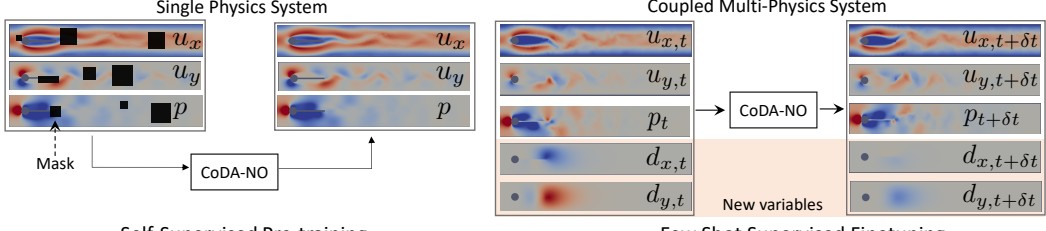

**Figure 1. CoDA-NO adapts seamlessly to new multi-physics systems.** Pre-trained on fluid dynamics data (Navier-Stokes equation with $u_x$, $u_y$, and $p$) using the masked-reconstruction objective, CoDA-NO easily adapts to multi-physics fluid-solid interaction systems (new $d_x$ and $d_y$ variables) without any architectural changes.

More generally, a foundation model trained on different kinds of PDEs can learn representations across multiple domains and then transfer them to new problems. Such foundation models have found immense success in computer vision and natural language processing [5, 6]. Foundation models are first trained in a self-supervised manner on large and often unlabeled datasets. Then they can be efficiently adapted or fine-tuned to a broad range of downstream tasks with minimal to no additional data or training.

Recent works have attempted to train a foundation model for solving PDEs [7–9]. However, these methods only work on predetermined PDEs with a fixed number of variables, and none of them consider multi-physics PDEs, and they are mostly restricted to uniform grids, limiting their applicability. For example, standard patching-based approaches used in Vision Transformers (ViTs) [10] often struggle with discontinuities in predicted functions and changing resolutions [11]. Since they are limited to fixed uniform grids, they cannot generalize to resolutions different from the training resolutions.

To handle varying resolutions and grids, *neural operators* [12, 13] have been introduced as a deep learning framework for learning mappings between function spaces. Neural operators are guaranteed to converge to a unique operator in the limit of increasingly fine discretizations (of the computational domain). This property is known as *discretization convergence*, making them agnostic to the discretization of the input and output functions and suitable for approximating solution operators of PDEs. Neural operators can replace numerical solvers while being significantly faster in several scenarios [14, 15]. While some of the previous PDE foundation models [7, 9] use neural operators, they still cannot handle multiphysics or coupled PDEs. They also cannot adapt to new variables that are not predetermined at the beginning of training.

**Our Approach:** We propose a novel *transformer* neural operator architecture with codomain attention (CoDA-NO) layers designed to handle varying combinations of physical phenomena modeled through coupled PDEs. We partition the input function codomain-wise into a set of token functions, each corresponding to distinct physical variables of the PDE. The CoDA-NO model processes this set of functions as input, extending the transformer architecture from a finite-dimensional vector space to an infinite-dimensional function space. This extension is achieved by carefully redesigning positional encodings, the self-attention mechanism, and normalization techniques.

In our architecture, each token is treated as a function, capturing cross-function dynamics through attention mechanisms while maintaining *discretization convergence*. This design empowers the architecture to handle functions discretized on grids of varying resolutions. Specifically, each token function is subjected to the following operations: (i) concatenation with a learned positional embedding, (ii) lifting to a higher-dimensional co-domain, and (iii) functional attention mechanisms to compute interactions. We use Fourier neural operators (FNOs) [16] rather than traditional multi-layer perceptrons (MLPs) to create the representations for keys, values, and queries, which helps maintain the functional nature of the input data. Details can be found in Sec. 3 and Alg. 1.

CoDA-NO can be applied to varying numbers of input functions (on different geometries) and adapt to novel PDEs with fewer or additional interacting variables, as illustrated in Fig. 1. This allows us to learn multiple PDE systems in one model.

To demonstrate CoDA-NO's generalizability across diverse physical systems, we examine two settings: multiphysics problems and a collection of single-physics problems.

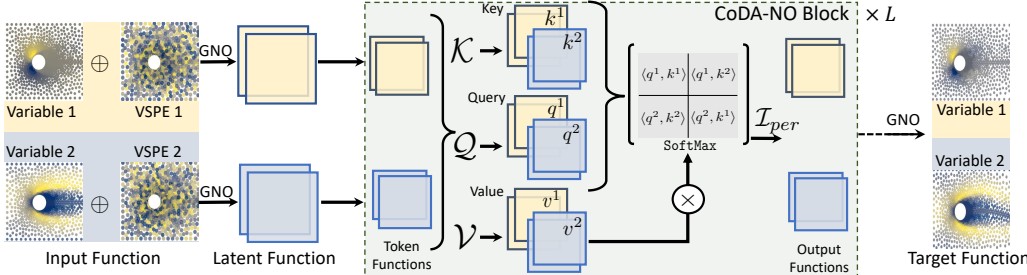

**(a) Illustration of CoDA-NO architecture.** Each of the input physical variables, 'variable 1' and 'variable 2', depicted with two different colors (yellow and blue), incorporate a learnable variable-specific positional encoder (VSPE). These variables, along with the corresponding VSPE, are passed through GNO layers to transform from non-uniform to latent uniform grids. Codomain attention tokenizes the latent functions along the codomain. Each token undergoes transformations with $\mathcal{K}$, $\mathcal{Q}$, and $\mathcal{V}$ operators yielding key, query, and value functions $\{k^1, k^2\}$, $\{q^1, q^2\}$, and $\{v^1, v^2\}$. The resulting function is computed using a self-attention mechanism in function space followed by an integral operator $\mathcal{I}_{per}$. Finally, the output function on the target geometry is generated by passing through stacked CoDA-NO blocks, followed by an additional GNO layer.

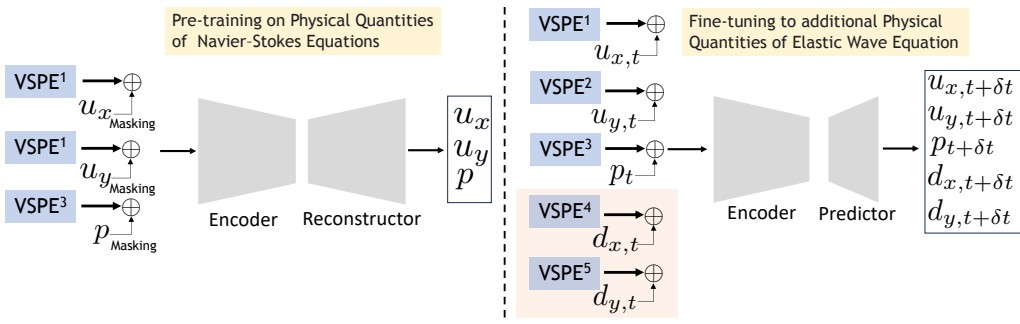

**(b) Self-supervised pre-training and fine-tuning with CoDA-NO.** Model, pre-trained on the Navier-Stokes equations dataset (with $u_x$, $u_y$, and $p$) in a self-supervised way, can be fine-tuned to a fluid-solid interaction dataset (new $d_x$ and $d_y$ variables) by only including two extra VSPEs and a predictor module.

**Figure 2.** (a) CoDA-NO architecture. (b) Self-supervised pre-training and fine-tuning process with CoDA-NO.

For the multiphysics scenario, we examine two distinct systems. First, we consider a fluid-structure interaction problem [17] governed by the incompressible Navier-Stokes equation and the Elastic wave equation. The fluid-structure interaction problem is representative of the multi-physics behavior of various real-world problems, e.g., climate and atmosphere modeling. It also provides an additional challenge of irregular meshes on a complex geometry.

Instead of directly learning to solve the full multiphysics problem, we start with a curriculum where we first learn the basic fluid dynamics without the elastic wave equation, governed by the incompressible Navier-Stokes equation, with velocity and pressure as variables. We pre-train CoDA-NO in a self-supervised manner on snapshots of fluid flows by masking different parts of the velocity or pressure fields. Using few-shot supervised fine-tuning, we show that our model can adapt to unseen viscosities and additional displacement fields given by the elastic wave equation. We use graph neural operator (GNO) layers [18] as encoders and decoders to handle time-varying irregular meshes of the fluid-structure interaction problems. For the few-shot learning problem, our model achieves 36.8% lower errors on average compared to the best-performing baseline trained from scratch on the target problem.

The second system involves Rayleigh-Bénard convection, where the Navier-Stokes and heat (energy) equations are coupled in a regular $2D$ domain. Similar to the first case, we pre-train CoDA-NO with an incompressible Navier-Stocks equation involving just the velocity term. Then, we fine-tuned the model to predict velocity and temperature using few-shot training samples. Here, too, the pre-trained CoDA-NO significantly outperforms the baseline, reducing the prediction error by a factor of two.

We also train CoDA-NO on a diverse set of PDEs, which form a subset of PDEBench [19] and demonstrate superior performance and parameter efficiency over prior approaches in learning all of those PDE systems. CoDA-NO consistently outperforms the FNO architecture trained on the same set of PDEs, reducing test error by up to 43% while only requiring 2% of the parameters.

**Our contributions are as follows:**

- We propose a co-domain attention neural operator that efficiently learns solution operators to PDEs by formulating transformer operations in function space and ensuring discretization convergence.
- The proposed architecture enables self-supervised learning in function space for diverse physical systems by handling varying numbers of input functions and geometries.
- CoDA-NO achieves state-of-the-art performance in generalizing to unknown physical systems with very limited data. That is, CoDA-NO can be viewed as the first foundation neural operator for multiphysics problems.

## 2   Related Works

**Transformers for solving PDEs.** Recent work [20] proposes a method to weight variables/codomains of the input function based on the weights calculated from the PDE parameters. Another study [21] proposes a scalable transformer architecture by combining a projection operator to a one-dimensional domain and a learnable factorized kernel. In contrast to these works, CoDA-NO provides a complete attention operator by considering each physical variable as a token function, i.e., an infinite-dimensional vector, extending traditional transformers that only operate on finite-dimensional tokens.

**Self-supervised learning.** Self-supervised learning (SSL) has been proposed to tackle the issue of limited labeled data [22–24]. It allows the training of large *foundation models* on massive amounts of unlabeled data in the field of computer vision and natural language processing. Subsequently, these models can be successfully applied to a wide range of downstream tasks with minimal to no additional task-specific data [6, 25–27].

**Pre-training for PDE solving.** Models that are pre-trained in a self-supervised fashion have also gained traction in the domain of scientific computing. One recent study [8] proposes pretraining the models with autoregressive tasks on a diverse dataset of multiple PDEs. These models can then be fine-tuned for specific downstream PDEs. Several recent studies have investigated task-agnostic approaches through masking-and-reconstruction [22] and the consistency of representations under symmetry transformations [16, 28, 29]. Recent work [7] also sheds light on the transferability of these models between different systems of PDEs. While these methods achieve good performance, the target (downstream) PDE must maintain a strict resemblance to the ones used for pretraining. In addition, adapting these models for PDEs with new additional physical variables is not possible. Additionally, ViT-based patching approaches [11] disrupt the continuity and are not resolution-agnostic.

## 3   Method

Let us first define our setting and provide a brief introduction to neural operators. For further details, we refer to Sec. A in the appendix.

For an input function $a\colon \mathcal{D} \to \mathbb{R}^{d_{in}}$, we will denote the $d_{in}$-dimensional output space $\mathbb{R}^{d_{in}}$ as the *codomain*. We consider the components of the codomain as different physical variables, given by real-valued functions over the input domain $\mathcal{D}$, i.e., $a = [a^1, \ldots, a^{d_{in}}]$ with $a^i : \mathcal{D} \to \mathbb{R}$. The same applies to the output function $u\colon \mathcal{D} \to \mathbb{R}^{d_{out}}$. We define the action of a *pointwise operator* $\mathcal{H} : \{f : \mathcal{D} \to \mathbb{R}^{d_f}\} \to \{g : \mathcal{D} \to \mathbb{R}^{d_g}\}$ given by a function $h_\theta : \mathbb{R}^{d_f} \to \mathbb{R}^{d_g}$ with parameters $\theta$ as

$$\mathcal{H}[f](x) = h_\theta(f(x)). \tag{1}$$

Moreover, we define an *integral operator* $\mathcal{T} : \{f : \mathcal{D} \to \mathbb{R}^{d_f}\} \to \{g : \mathcal{D} \to \mathbb{R}^{d_g}\}$ given by a kernel function $k_\phi$ with parameters $\phi$ as

$$\mathcal{T}[f](x) = \int_{\mathcal{D}} k_\phi(x, y) f(y) \, \mathrm{d}y. \tag{2}$$

**Problem Statement.** Our objective is to construct a general neural operator architecture that explicitly represents the interaction between the physical variables of PDE systems. Such an architecture should be able to learn and predict various systems without being constrained to a fixed number of variables.

Let's consider two input functions $a\colon \mathcal{D} \to \mathbb{R}^{d_{in}}$ and $\tilde{a}\colon \mathcal{D} \to \mathbb{R}^{\tilde{d}_{in}}$ of two different PDE with corresponding output functions $u\colon \mathcal{D} \to \mathbb{R}^{d_{out}}$ and $\tilde{u}\colon \mathcal{D} \to \mathbb{R}^{\tilde{d}_{out}}$. In general, the functions $a$ and $\tilde{a}$

represent $d_{in}$ and $\tilde{d}_{in}$ physical variables over the domain $\mathcal{D}$ with $d_{in} \neq \tilde{d}_{in}$ . We aim to design neural operator architectures $\mathcal{G}$ that can both be applied to $a$ as well as $\tilde{a}$ despite the different codomains of the input as well as output functions.

Such property provides the possibility to evaluate or finetune the operator on PDEs with different numbers of variables than those on which it was trained. In particular, when the PDE systems have overlapping physical variables $\{a^i\}_{i=1}^{d_{in}} \cap \{\tilde{a}^i\}_{i=1}^{\tilde{d}_{in}} \neq \emptyset$, this naturally allows to transfer learned knowledge from one system to the other. We will next describe the details of the CoDA-NO layers and architecture to achieve this goal.

**Neural Operator on Sets.** As we consider the vector-valued input function $a$ as a set of $d_{in}$ functions $\{a^1, a^2, \ldots, a^{d_{in}}\}$ that represents different physical variables of the PDE, we seek to construct operators that act on *sets* of input functions with different cardinalities.

For an efficient implementation of operators on sets of functions, we mimic transformer architectures and share weights across different variables. Specifically, we can define the integral operator $\mathcal{I}_{per}$ as

$$\mathcal{I}_{per}[a] = \left[ \mathcal{I}[a^1], \ldots, \mathcal{I}[a^{d_{in}}] \right], \tag{3}$$

where $a = [a^1, \ldots, a^{d_{in}}]$ and $\mathcal{I}$ is a regular integral operator as described in Eq. (2). Such construction makes the operator *permutation-equivariant* with respect to the order of the variables in the set. Following the same mechanism, we can also define permutation-equivariant pointwise operator $\mathcal{H}_{per}$ with a shared pointwise operator $\mathcal{H}$ (see Eq. (1)). We will use FNO$_{per}$ and GNO$_{per}$ to denote permutation-equivariant operators using a shared GNO and FNO, respectively.

**CoDA-NO Layer.** To explain the CoDA-NO layer, let us assume the input function $a$ has been processed into a latent function $w : \mathcal{D} \to \mathbb{R}^d$. We partition the function into a set of so-called *token functions* $w^j : \mathcal{D} \to \mathbb{R}^{d'}$ with $w^j \in \mathcal{W}$ for $j \in \{1, \ldots T\}$ along the codomain, such that $w = \left[ w^1, \ldots w^T \right]$ (and where each $w^j$ is associated with precisely one of the physical input variables). That is, $w$ represents the codomain-wise concatenation of the token functions $w^j$ and $d' = \frac{d}{T}$. If no other value is specified, we assume that $d' = 1$. The CoDA-NO layer now processes the token functions using an extension of the self-attention mechanism to the function space (see Appendix Sec. B and Fig. 2a).

Let us begin by introducing a single-head CoDA-NO layer. Later, we will expand the concept to multi-head codomain attention. We extend the key, query, and value *matrices* of the standard attention (see Appendix Sec. B for details) to *operators* mapping token functions $w^j : \mathcal{D} \to \mathbb{R}^{d'}$ to key, query, and value functions. We define the key, query, and value operators as

$$\mathcal{K} : \mathcal{W} \to \{k^j : \mathcal{D} \to \mathbb{R}^{d_k}\}, \quad \mathcal{Q} : \mathcal{W} \to \{q^j : \mathcal{D} \to \mathbb{R}^{d_q}\}, \quad \mathcal{V} : \mathcal{W} \to \{v^j : \mathcal{D} \to \mathbb{R}^{d_v}\}. \tag{4}$$

Assuming $d_k = d_q$, we denote by $k^j = \mathcal{K}[w^j]$, $q^j = \mathcal{Q}[w^j]$, and $v^j = \mathcal{V}[w^j]$ the key, query, and value functions of the token functions, respectively.

Next, we calculate the output (token) functions $o^j : \mathcal{D} \to \mathbb{R}^{d_v}$ as

$$o^j = \texttt{Softmax} \left( \begin{bmatrix} \frac{\langle q^j, k^1 \rangle}{\tau} \\ \vdots \\ \frac{\langle q^j, k^T \rangle}{\tau} \end{bmatrix} \right) [v^1, \ldots, v^T]^\top, \tag{5}$$

where $\tau$ is the *temperature* hyperparameter. Here, $\langle ., . \rangle$ denotes a suitable dot product in the function space. We take the $L^2(\mathcal{D}, \mathbb{R}^{d_k})$-dot product given by $\langle q^j, k^m \rangle = \int_{\mathcal{D}} \langle q^j(x), k^m(x) \rangle \, dx$, where the integral can be discretized using quadrature rules, similar to the integral operator in Eq. (2).

To implement multi-head attention, we apply the (single-head) attention mechanism described above separately for multiple heads $h \in \{1, \ldots H\}$ using $\mathcal{K}^h, \mathcal{Q}^h$, and $\mathcal{V}^h$ to obtain $o^{j,h}$. We then concatenate these outputs $o^{j,h}$ along the codomain and get $c^j := [o^{j,1}, \ldots o^{j,H}]$. Finally, we use an operator

$$\mathcal{M} : \{c^j : \mathcal{D} \to \mathbb{R}^{H \cdot d_v}\} \to \{o^j : \mathcal{D} \to \mathbb{R}^{d_v}\} \tag{6}$$

to get the output function $o^j$.

We obtain the output of the attention mechanism by concatenating $o^j$s as $o = [o^1, o^2, \ldots o^T]$. Finally, we complete the CoDA-NO layer by applying a permutation-equivariant integral operator $\mathcal{I}_{per}$ on $o$. When CoDA-NO is acting on functions sampled on a uniform grid, the internal operators $\mathcal{K}^h, \mathcal{Q}^h, \mathcal{V}^h, \mathcal{M}$, and $\mathcal{I}$ are implemented as FNOs.

**Function Space Normalization.** Normalization is a vital aspect of deep learning architectures. However, when it comes to neural operators mapping infinite-dimensional functions, this topic remains largely unexplored. We now provide a natural extension. Given a function $w$, let $w^j : \mathcal{D} \to \mathbb{R}^{d'}$ be a token. Then we calculate the mean $\mu \in \mathbb{R}^{d'}$ and standard deviation $\sigma \in \mathbb{R}^{d'}$ for this token as

$$\mu^j = \int_{\mathcal{D}} w^j(x)\,\mathrm{d}x, \quad \sigma^j = \left( \int_{\mathcal{D}} (w^j(x) - \mu^j)^{\circ 2}\,\mathrm{d}x \right)^{\circ \frac{1}{2}}. \tag{7}$$

Here, $\circ r$ denotes the elementwise (Hadamard) $r^{th}$-power. The normalization operator can be written as

$$\texttt{Norm}[w^j](x) = (\mathbf{g} \oslash \sigma^j) \odot (w^j(x) - \mu^j) + \mathbf{b}.$$

Here $\mathbf{b} \in \mathbb{R}^{d'}$ and $\mathbf{g} \in \mathbb{R}^{d'}$ are learnable bias and gain vectors and $\oslash$ and $\odot$ denote elementwise division and multiplication operation. This normalization can be seen as an extension of *instance normalization* [30] for function spaces. Similarly, normalization variants, such as *group norm*, *layer norm*, and *batch norm*, extend to operator learning with these definitions of statistics [31–33].

**Variable Specific Positional Encoding (VSPE).** We learn positional encoders $e^i : \mathcal{D} \to \mathbb{R}^{d_{en}}$ for each physical variable $i \in \{1, \ldots, d_{in}\}$, for the given vector-valued input function $a = [a^1, \ldots, a^{d_{in}}]$. We concatenate each positional encoding $e^i$ with the respective variable $a^i : \mathcal{D} \to \mathbb{R}$ along the codomain to obtain extended input functions $\bar{a}^i = [a^i, e^i]$. Next, we apply a shared pointwise lifting operator $\mathcal{P} : \{\bar{a}^i : \mathcal{D} \to \mathbb{R}^{d_{en}+1}\} \to \{\bar{w}^i : \mathcal{D} \to \mathbb{R}^D\}$, typically with $D > d_{en} + 1$. Finally, we concatenate $\bar{w}^i$, $i \in \{1, \ldots d_{in}\}$, to get the lifted latent function

$$w = [\bar{w}^1, \ldots, \bar{w}^{d_{in}}] : \mathcal{D} \to \mathbb{R}^{D \cdot d_{in}}. \tag{8}$$

In the previous paragraphs, we used $d = D \cdot d_{in}$ and, to maintain the permutation-equivariance property of the operator, $d'$ must divide $D$.

Algorithm 1 presents the pseudocode for the CoDA-NO architecture applied to input functions $[a^1, a^2]$, mapping two different physical variables on a uniform grid in a 1D domain, to the solution functions $[u^1, u^2]$. It assumes $d' = D$ while designing the CoDA-NO layer. Notably, to incorporate another function $a^3$, representing a new physical variable, it is only necessary to introduce a corresponding parameter for the new VSPE, denoted as $\kappa^3$.

To effectively handle non-uniform complex geometries, we follow the GINO architecture

---

**Algorithm 1** Adaptation of CoDA-NO from two physical variables $a^1, a^2$ to a new variable $a^3$ on uniform $1D$ grid. Only the parts in 'Blue' are additionally required to incorporate the new variable $a^3$.

**Require:** $a^1, a^2, a^3 \in \mathbb{R}^{1 \times n}$
**Variable Specific Positional Encoding:**
```
// Learnable Fourier coefficients
κ¹, κ², κ³ ∈ ℂ^{d_en × n/2}
ā¹ ← concatenate(a¹, iFFT(κ¹))
ā² ← concatenate(a², iFFT(κ²))
ā³ ← concatenate(a³, iFFT(κ³))
```

**Lifting:**
```
// Lifting from ℝ^{d_en+1×n} to
ℝ^{D×n} with D > d_en + 1
w¹ ← pointwiseMLP(ā¹)
w² ← pointwiseMLP(ā²)
w³ ← pointwiseMLP(ā³)
```

**Codomain Attention Block:** $\times L$
```
k¹, k², k³ ← FNO_k(w¹), FNO_k(w²), FNO_k(w³)
q¹, q², q³ ← FNO_q(w¹), FNO_q(w²), FNO_q(w³)
v¹, v², v³ ← FNO_v(w¹), FNO_v(w²), FNO_v(w³)
// Compute attention coefficients M
for i, j ∈ {1, 2, 3} do
    M[i, j] ← ⟨q^i, k^j⟩
end for
M ← softmax( M[i,j] / N )
// Compute attention outputs
for i ∈ {1, 2, 3} do
    o^i ← ∑_{j∈{1,2,3}} v^j × M[i, j]
end for
// Normalize and Residual
o¹ ← norm(o¹) + w¹
o² ← norm(o²) + w²
o³ ← norm(o³) + w³
w¹, w², w³ ← FNO_I(o¹), FNO_I(o²), FNO_I(o³)
⋮
```

**Projection:**
```
// Projection from ℝ^{D×n} to ℝ^{1×n}
u¹ ← pointwiseMLP(w¹)
u² ← pointwiseMLP(w²)
u³ ← pointwiseMLP(w³)
return u¹, u², u³
```

[34], where a GNO is used as an encoding and decoding module. Given a set of evaluations of an input function $a$ on a mesh, as represented by $\{a(x_i^{in})\}_{i=1}^n$, where $\{x_i^{in}\}_{i=1}^n \subset \mathcal{D}_{in}$, our first step involves concatenation of each physical variables with respective VSPEs (see Fig. 2a).

Next, we use $\text{GNO}_{per}$ to transform the function $a$ into a new function $w_0$ on a uniform latent grid, represented by $\{x_i^{grid}\}_{i=1}^{n'}$. Finally, we apply $l$ stacked CoDA-NO layers to $w_0$ to obtain the encoded function $w_l$, which acts as a representation of the input function $a$.

The decoding module is essentially a mirrored version of the encoding module. It starts by applying another block of $l$ stacked CoDA-NO layers to the encoded function $w_l$ to obtain $w_L$. Subsequently, it uses another $\text{GNO}_{per}$ operator to transform $w_L$ on a uniform grid to an approximation $u$ of the solution function on an arbitrary output grid $\{u(x_i^{out})\}_{i=1}^{n'}$. The architecture is summarized in Fig. 2a.

**Model Training.** To seamlessly adapt to multi-physics PDEs with limited data, we propose a two-stage training process: Self-supervised pretraining is followed by a supervised fine-tuning stage. For a summary, we refer to Fig. 2b.

***Pre-training.*** In the context of self-supervised pretraining, the objective is to train the model to reconstruct the original input function from its masked version. Within this phase, the model's encoding component is denoted as the *Encoder*, while the decoding component comprises the *Reconstructor*. The values of the input function at a specific percentage of mesh points are randomly masked to zero, and certain variables (channels/co-domains) of the input function are entirely masked to zero. The model is then trained to reconstruct the original input from this masked version.

We emphasize that the self-supervised learning phase is agnostic of the downstream supervised task and only requires snapshots of simulations of the physical systems.

***Fine-tuning.*** In the supervised fine-tuning phase, the *Reconstructor* is omitted from the decoding module and replaced by a randomly initialized Predictor module. The parameters of the Encoder and VSPEs are copied from pre-trained weights. If the fine-tuning (target) PDE introduces variables that are not present in the pre-training PDE; we train additional variable encoders only for these newly introduced variables (see Fig. 2b). This ensures that the model adapts to the expanded set of variables needed for the fine-tuning task with minimal additional parameters.

## 4 Experiments

We conduct experiments on two coupled PDEs: fluid-structure interaction and Rayleigh-Bénard convection system. We also test our model on a diverse set of PDEs from PDEBench [19]. The code is available at `https://github.com/neuraloperator/CoDA-NO`.

**Modeling Fluid-Structure Interaction.** We consider the following problems: (a) a fluid dynamics problem, where a Newtonian, incompressible fluid impinges on a rigid object, and (b) a fluid-structure interaction problem between a Newtonian, incompressible fluid and an elastic, compressible solid object [17]. We denote $\Omega_t^f$ (resp. $\Omega_t^s$) as the domain occupied by the fluid (resp. the solid) at time $t$. The dynamics of the fluid are governed by the Navier-Stokes equations

$$\rho^f \frac{\partial u}{\partial t} + \rho^f \nabla \cdot (u \otimes u) = \nabla \cdot \boldsymbol{\sigma}^f, \ \nabla \cdot u = 0, \quad \text{in } \Omega_t^f \tag{9}$$

where $u$ and $\rho^f$ denote the fluid velocity and density, respectively. And $\boldsymbol{\sigma}^f$ denotes the Cauchy stress tensor, given by $\boldsymbol{\sigma}^f = -p\mathbb{I} + \mu(\nabla u + \nabla u^T)$, where $\mathbb{I}$ is the identity tensor, $p$ the fluid pressure, and $\mu$ the fluid dynamic viscosity.

For fluid-structure interaction, the deformable solid is governed by the elastodynamics equations

$$\rho^s \frac{\partial^2 d}{\partial t^2} = \nabla.(J\boldsymbol{\sigma}^s(\mathbf{F}^{-1})^T) \qquad \text{in } \Omega_t^s \tag{10}$$

with $\mathbf{F} = \mathbb{I} + \nabla d$ and $J = \det(\mathbf{F})$. Here $d$, $\rho^s$, $F$, and $\boldsymbol{\sigma}^s$ denote the deformation field, the solid density, the deformation gradient tensor, and the Cauchy stress tensor, respectively (see Eq. (18) in the Appendix). The fluid dynamics (resp. the fluid-structure interaction) problem considers a fluid flow past a fixed, rigid cylinder with a rigid (resp. elastic) strap attached. The details regarding the geometric setup (see Fig. 3), time-dependent inlet boundary condition, and the initial conditions are

provided in the Appendix Sec. C.1.

**Modeling Rayleigh-Bénard Convection.** The Rayleigh-Bénard convection system governs the flow of a fluid layer heated from below and cooled from above. The governing equations for the Rayleigh-Bénard system consist of the incompressible Navier-Stokes equations coupled with an energy equation for heat transfer. The system is modeled as follows:

$$\frac{\partial \mathbf{u}}{\partial t} + \mathbf{u} \cdot \nabla \mathbf{u} + \nabla P - \nu \nabla^2 \mathbf{u} - \alpha g \mathbf{T}\hat{\mathbf{z}} = 0 \tag{11}$$

$$\frac{\partial T}{\partial t} + \mathbf{u} \cdot \nabla \mathbf{T} - \kappa \nabla^2 \mathbf{T} = 0 \tag{12}$$

**Dataset Description and Generation.** To study the fluid-structure interaction system, two datasets, the fluid-structure interaction (NS+EW dataset) and the fluid dynamics(NS dataset), are generated using the TurtleFSI package [35].

We simulate the fluid-structure interaction and the fluid dynamics test cases described above up to time $T_f = 10$, using a constant time-step $\delta t = \frac{T_f}{n}$, where $n = 1000$. The data sets are composed of solution trajectories $[u_t, p_t, d_t]$ (resp. $[u_t, p_t]$), which denote the approximate solution of the fluid-structure interaction problem (resp. the fluid dynamics problem) at times $t = i\delta t, i \in \{0, \dots, n\}$. These trajectories are generated on the basis 3 parameters $(\mu, c_1, c_2)$ describing combinations of fluid viscosities $\mu \in \{0.5, 1, 5, 10\}$ and inlet conditions, $(c_1, c_2) \in \mathcal{I}$.

For our setup, the fluid considered is water, with a density of $1000 kg.m^{-3}$ and a maximum inlet velocity of approximately $4 m.s^{-1}$, leading to Reynolds ($Re$) numbers in the range $200 - 4000$ (for $\mu$ between $10 - 0.5$). Modeling fluid-solid interaction or only fluid motion with such high Reynolds numbers is challenging and serves as a benchmark problem [12, 17] (See Sec. C.2 for a detailed explanation).

To study the Rayleigh-Bénard convention system, we degenerate two different PDE datasets. Firstly, we generate Rayleigh-Bénard convection system with $Ra$ number $12 \times 10^3$ and $20 \times 10^3$. We set the temperature difference between the top (cold) and bottom (hot) boundaries to 1. We assume no-slip boundary conditions, and to start the convection process, we also add initial temperature perturbation. Additionally, we generate incompressible Navier-Stocks equations with Reynold number $Re = 500$ with cyclic boundary condition on a uniform $2D$ grid [36] (for details, see Appendix Sec. C.3).

**Experiment Setup.** For the fluid-structure interaction system, we conduct two distinct pretraining procedures for CoDA-NO and obtain two pretrained models: $\mathcal{G}_{\text{NS}-\text{EW}}^{\text{p}}$ and $\mathcal{G}_{\text{NS}}^{\text{p}}$. The former is pretrained on a fluid-structure interaction dataset that combines the Navier-Stokes equation and the elastic wave equation, denoted as $\mathcal{G}_{\text{NS}-\text{EW}}^{\text{p}}$. The latter, $\mathcal{G}_{\text{NS}}^{\text{p}}$, is pretrained on a fluid motion dataset governed solely by the Navier-Stokes equation. In both scenarios, the pretraining involves utilizing 8000 snapshots of flow and displacement fields with $Re \in \{200, 2000\}$.

The supervised task involves training the model to predict the system's state at the subsequent time step based on its current state. For the fluid-structure interaction dataset, we train an operator $\mathcal{G}_{\text{NS}-\text{EW}}$ such that $\mathcal{G}_{\text{NS}-\text{EW}} : [u_t, p_t, d_t] \rightarrow [u_{t+\delta t}, p_{t+\delta t}, d_{t+\delta t}]$, where $u, p,$ and $d$ are the velocity, pressure, and mesh deformation fields (see Sec. 4). For the data with only fluid motion, we train the operator $\mathcal{G}_{\text{NS}}$ which maps between the current and next time step velocity and pressure field as $\mathcal{G}_{\text{NS}} : [u_t, p_t] \rightarrow [u_{t+\delta t}, p_{t+\delta t}]$.

The pretrained model for both datasets is fine-tuned for unseen viscosity $\mu = 5.0 (Re = 400)$ with different numbers of a few shot examples. The inlet conditions of these simulations are excluded from the pretraining data. So, the target PDEs' viscosity and inlet conditions are absent in the per-taining dataset. We test the model's adaptability on a more turbulent fluid-solid interaction dataset with $Re = 4000 (\mu = 0.5)$ by finetuning both pretrained models $\mathcal{G}_{\text{NS}-\text{EW}}^{\text{p}}$ and $\mathcal{G}_{\text{NS}}^{\text{p}}$ on each dataset.

For the Rayleigh-Bénard convention system, we pretrain a CoDA-NO model, denoted as $\mathcal{G}_{\text{NS}}^{\text{p}}$, on the incompressible Navier-Stokes equations using $40,000$ snapshots in a self-supervised manner. The supervised task for this system is to train an operator, $\mathcal{G}_{\text{NS}-\text{T}} : [u_t, \mathbf{T}_t] \rightarrow [u_{t+\delta t}, \mathbf{T}_{t+\delta t}]$, where $u$ represents velocity and $T$ represents temperature. The pretrained model $\mathcal{G}_{\text{NS}}^{\text{p}}$ is fine-tuned for the

**Table 1.** Test $L_2$ loss for fluid dynamics (NS) and fluid-solid interaction (NS+EW) datasets with viscosity $Re = 400$ and $Re = 4000$ for different numbers of few-shot training samples.

| Model | Pretrain Dataset | $Re = 400$ | | | | | | $Re = 4000$ | | |
|---|---|---|---|---|---|---|---|---|---|---|
| | | # Few Shot Training Samples | | | | | | | | |
| | | 5 | | 25 | | 100 | | 5 | 25 | 100 |
| | | Evaluation Dataset | | | | | | | | |
| | | NS | NS+EW | NS | NS+EW | NS | NS+EW | NS+EW | NS+EW | NS+EW |
| GINO | - | 0.200 | 0.122 | 0.047 | 0.053 | 0.022 | 0.043 | 0.717 | 0.292 | 0.136 |
| DeepO | - | 0.686 | 0.482 | 0.259 | 0.198 | 0.107 | 0.107 | 0.889 | 0.545 | 0.259 |
| GNN | - | 0.038 | 0.045 | 0.008 | 0.009 | 0.008 | 0.009 | 0.374 | 0.310 | 0.132 |
| ViT | - | 0.271 | 0.211 | 0.061 | 0.113 | 0.017 | 0.021 | 0.878 | 0.409 | 0.164 |
| U-Net | - | 13.33 | 3.579 | 0.565 | 0.842 | 0.141 | 0.203 | 3.256 | 0.563 | 0.292 |
| | - | 0.182 | 0.051 | 0.008 | 0.084 | 0.006 | 0.004 | 0.326 | 0.264 | 0.070 |
| Ours | NS | 0.025 | 0.071 | 0.007 | 0.008 | **0.004** | 0.005 | 0.366 | 0.161 | 0.079 |
| | NS+EW | **0.024** | **0.040** | **0.006** | **0.005** | 0.005 | **0.003** | **0.308** | **0.143** | **0.069** |

supervised task of solving Rayleigh-Bénard convection using different numbers of a few shot training samples.

**Baselines.** For comparison on the supervised tasks on fluid-structure interaction system, we train GINO [18], DeepONet [37], graph neural network (GNN) [38], vision transformer (ViT) [10], and the Unet [39] model from scratch. The mesh points of the NS and NS+EW datasets are irregular and change for each sample. So, to efficiently handle irregular mesh, in the *branch* network of DeepONet, we use a GNN layer followed by MLPs. Also, as ViT and Unet can handle irregular mesh, we follow the architecture of GINO and use a GNN layer to query the latent function on a uniform grid. We then apply Unet and ViT to the uniform grid, followed by another GNN layer, to get the output at the desired query points. For the Rayleigh-Bénard convection system, we train Unet [39] and FNO [12] from scratch and compare them against our proposed model.

It should be noted that employing the existing models for pertaining and subsequent finetuning on the target datasets is nontrivial due to complex geometry and the changes in the number of physical variables between the pertaining and target datasets. We report the $L^2$ error between the predicted and target functions, which serves as a measure of model performance. Additional implementation details are provided in the Appendix Sec. H.

**Results.** In Tab. 1, we report the performance of our model and the baselines for modeling the fluid-structure interaction. We observe that the pretrained CoDA-NO model performs better than the baselines. Importantly, the performance gain is higher when the number of few-shot examples is very low. This demonstrates the sample efficiency and generalization capability of CoDA-NO to previously unseen physical systems.

Next, when CoDA-NO is pretrained solely on the NS dataset, it shows an impressive ability to adapt to the more challenging NS+EW dataset. Finally, when CoDA-NO is pretrained on the more intricate NS+EW dataset, it easily adapts to the simpler NS dataset through fine-tuning. This underscores the capability of the CoDA-NO to adjust between different PDEs with varying numbers of variables seamlessly.

Also, we notice that pretrained CoDA-NO performs better than CoDA-NO trained from scratch, demonstrating the effectiveness of the pretraining scheme. We also provide the energy spectra of the predicted fluid flow by the different models in Sec. F.4 where we observe that the energy spectrum remains closest to the ground truth.

In Tab. 2, we present the result on modeling the Rayleigh-Bénard convention. We observe that pretrained CoDA-NO outperforms every other baseline and adapted to the new temperature variable, $T$, of the Rayleigh-Bénard system. Similar to the fluid-structure interaction problem, we also observe that the pretrained CoDA-NO outperforms CoDA-NO trained from scratch, which underlines the effectiveness of our pretraining and adaptation mechanism.

**Table 2.** Test $L_2$ error for Rayleigh-Bénard convection system with coupled Navier-Stokes and energy (heat) equation with Rayleigh number $Ra = 12 \times 10^3$ and $Ra = 20 \times 10^3$ for different few shot examples.

| Model | Pretrain dataset | $Ra = 12 \times 10^3$ | | | $Ra = 20 \times 10^3$ | | |
| | | #Few Shot Training Samples | | | | | |
| | | 5 | 10 | 25 | 5 | 10 | 25 |
| Unet | - | 0.049 | 0.025 | 0.013 | 0.126 | 0.083 | 0.075 |
| FNO | - | 0.119 | 0.070 | 0.044 | 0.491 | 0.166 | 0.127 |
| Ours | - | 0.067 | 0.045 | 0.035 | 0.221 | 0.058 | 0.040 |
| | NS | **0.016** | **0.007** | **0.002** | **0.074** | **0.040** | **0.029** |

Additionally, we also conduct experiments on various PDEs from the PDEBench dataset [19], where we show superior performance and parameter efficiency (see Appendix Sec. G).

**Adaptation to More Turbulent Fluid-Structure Interaction.** We also test the adaptation capability of our pretrained model on a more turbulent fluid-solid interaction scenario with viscosity $\mu = 0.5$ with a Reynolds number of 4000. From Tab. 1, we can observe that, even though the model is pretrained on data with lower Reynold's number $(200 - 2000)$, it can seamlessly adapt to more turbulent flow and outperform baselines with a significant margin.

**Ablation Studies.** To demonstrate the effect of each of the proposed components, namely, codomain attention, normalization layer, VSPE, and pertaining, we present the result of a detailed ablation study in Appendix Sec. F.1. We observe that substituting the codomain attention with regular patch-based attention impacts the model's performance. In particular, removing the normalization layer prevents the model from converging.

We also provide an ablation study on fine-tuning methods. Instead of fine-tuning all the parameters, here, we freeze the parameters of the "Encoder" and only train the parameters of the "Predictor" and VSPEs. This minimized the number of trainable parameters during fine-tuning. Also, in this case, we performed significantly better than the other models (see Appendix Sec. F.5).

We also provide the results for the zero-shot super-resolution task, where we directly predict the output function on a much denser mesh than the training mesh. Our findings show that CoDA-NO outperforms other baselines significantly (see Appendix Sec. F.2).

Additionally, we have conducted a comparative analysis of the parameter count and computational cost for each model, which points to the overfitting problem of the baseline when learning complex multi-physics PDEs (see Appendix Sec. F.3).

**Limitations.** In general, CoDA-NO's performance on target PDEs is influenced by the number of training examples, and we highlight the potential for further enhancement through the integration of physics-informed approaches.

## 5 Conclusion

In this work, we introduce CoDA-NO, a versatile pre-trained model architecture designed for seamless adaptation to Partial Differential Equations (PDEs) featuring diverse variable compositions. Departing from conventional patch-based attention modules, CoDA-NO innovatively extends the transformer to function spaces by computing attention across co-domains. Leveraging a flexible variable encoding scheme and a graph-based neural operator module, CoDA-NO exhibits adaptability to any target PDE, accommodating new and previously unseen variables with arbitrary input-output geometries during fine-tuning. Our empirical evaluations demonstrate that CoDA-NO consistently outperforms baselines across varying amounts of training data and exhibits robustness in handling missing variables. Our findings on complex multiphysics simulations underscore the efficacy and adaptability of CoDA-NO, positioning it as a valuable tool for addressing challenges in machine learning for PDEs.

## Acknowledgments

A. Anandkumar is supported in part by Bren endowed chair, ONR (MURI grant N00014-18-12624), and by the AI2050 senior fellow program at Schmidt Sciences. We thank David Pitt for his support in adding our code to the `neuraloperator` library, facilitating broader use and accessibility.

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

## Appendix

The appendix is organized as follows:

## A  Neural Operators

***Neural Operators*** are a class of deep learning architectures designed to learn maps between infinite-dimensional function spaces [40]. A Neural Operator seeks to approximate an operator $\mathcal{G}$ that maps an input function $a \in \mathcal{A}$ to its corresponding output function $u \in \mathcal{U}$ by building a parametric map $\mathcal{G}_\phi : \mathcal{A} \to \mathcal{U}$. The typical architecture of a Neural Operator can be described as $\mathcal{G}_\phi = \mathcal{P} \circ \mathcal{T}_L \circ \dots \mathcal{T}_1 \circ \mathcal{L}$. Here, $\mathcal{L}: a \to w_0$ and $\mathcal{P}: w_L \to u$ are lifting and pointwise projection operators, respectively. The action of any pointwise operator $\mathcal{H} : \{f : \mathcal{D} \to \mathbb{R}^{d_f}\} \to \{g : \mathcal{D} \to \mathbb{R}^{d_g}\}$ can be defined as

$$\mathcal{H}[f](x) = h_\theta(f(x)), \tag{13}$$

where $h_\theta : \mathbb{R}^{d_f} \to \mathbb{R}^{d_g}$ is any function with parameters $\theta$. The integral operator $\mathcal{T}_l : w_{l-1} \to w_l$ performs a kernel integration over the input function $w_{l-1}$ as

$$\mathcal{T}_l[w_{l-1}](x) = \int_{\mathcal{D}_{l-1}} k_l(x, y) w_{l-1}(y) \, \mathrm{d}y. \tag{14}$$

Here, $\mathcal{D}_{l-1}$ is the domain of the function $w_{l-1}$. In the case of Fourier Neural operators (FNO) [16], a convolution kernel, i.e., $k_l(x, y) = k_l(x - y)$ was used. By the convolution theorem, this enables the representation of an integral operator as a pointwise multiplication of the Fourier coefficients as follows $w_l = \mathcal{F}^{-1}(\mathcal{F}(k_l) \odot \mathcal{F}(w_{l-1}))$.

For the Graph neural operator (GNO) [41], a small neighborhood $B_r(x) \cap \mathcal{D}_{l-1}$ around the point $x$ is considered instead of integrating over the whole domain $\mathcal{D}_{l-1}$, such that Eq. (2) changes to

$$w_l(x) = \int_{B_r(x) \cap D_{l-1}} k_l(x, y) w_{l-1}(y) \, \mathrm{d}y. \tag{15}$$

Given a set of evaluations of the function $w_{l-1}$ on points $\{y_i\}_{i=1}^n \subset \mathcal{D}_{l-1}$, the kernel integral can be approximated by

$$w_l(x) \approx \sum_{y_i \in B_r(x)} k_l(x, y_i) w_{l-1}(y_i) q_i, \tag{16}$$

where $q_i \in \mathbb{R}$ are suitable quadrature weights [40]. The discretized kernel integral can be viewed as a message passing on graphs, where the neighborhood of each point $x$ consists of all points within radius $r$.

## B  Attention mechanism for finite-dimensional vectors

Given three sets of vectors, so-called queries $\{\mathbf{q}_i\}_{i=1}^{N_q}$, keys $\{\mathbf{k}_i\}_{i=1}^{N_k}$, and values $\{\mathbf{v}_i\}_{i=1}^{N_v}$ with $N_k = N_v$ and matching dimensions of queries and keys, attention mechanism calculates weighted sums of the value vectors. Specifically, the set of output vectors $\{\mathbf{o}_i\}_{i=1}^{N_q}$ can be expressed that

$$\mathbf{o}_i = \mathbf{a}^i [\mathbf{v}_1, \dots \mathbf{v}_{N_v}]^\top, \quad i = 1, \dots N_q, \tag{17}$$

where $\mathbf{a}^i = \mathtt{SoftMax}[\frac{\langle \mathbf{q}_i, \mathbf{k}_1 \rangle}{\tau}, \dots, \frac{\langle \mathbf{q}_i, \mathbf{k}_{N_k} \rangle}{\tau}]$ and $\tau$ is the temperature term. For the *self-attention* mechanism, the key, query, and value vectors are calculated from some input sequence $\{\mathbf{z}\}_{i=1}^L$ using the key, query, and value matrices $\mathbf{K}, \mathbf{Q}$, and $\mathbf{V}$ as

$$\mathbf{q}_i = \mathbf{Q}\mathbf{z}_i, \quad \mathbf{k}_i = \mathbf{K}\mathbf{z}_i, \quad \mathbf{v}_i = \mathbf{V}\mathbf{z}_i.$$

## C   Dataset Description

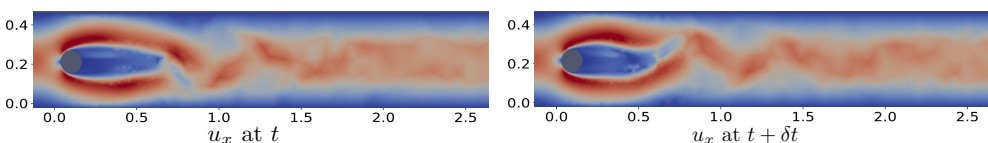

**Figure 3.** Visualization of **horizontal velocity** $u_x$ at $t$ and $t + \delta t$ time step.

### C.1   Fluid-Structure Interaction System

Here, we provide the details on generating the fluid-structure interaction dataset involving Navier-Stokes and Elastic wave equations.

**Fluid-structure interaction model.**   Under the Kirchoff St-Venant model, the Cauchy stress tensor $\boldsymbol{\sigma}^s$ verifies

$$\boldsymbol{\sigma}^s = \frac{1}{J}\mathbf{F}(\lambda^s(tr(\mathbf{E}))\mathbb{I} + 2\mu^s\mathbf{E})\mathbf{F}^T \tag{18}$$

where $\lambda^s$ and $\mu^s$ are the Lame coefficients, and

$$\mathbf{E} = \frac{1}{2}(\mathbf{F}\mathbf{F}^T - \mathbb{I}).$$

**Inlet Boundary Condition.**   Time-dependent inlet boundary conditions consist of $4^{th}$ order polynomials velocity profiles which vanish at the channel walls [42, 43]. The inlet conditions are given by

$$u^{\mathcal{I}}_{c_1,c_2}(y,t) = v(t) \cdot \frac{y(y - H)\left(y - c_1\frac{H}{2}\right)\left(y - c_2\frac{H}{2}\right)}{H(1 - c_1)(1 - c_2)}. \tag{19}$$

Here $v$ is the ramp function defined as

$$v(t) = \begin{cases} 70 \cdot \left(1 - \cos\left(\frac{\pi t}{2}\right)\right) & \text{if} \quad 0 \leq t < 2 \\ 140 & \text{if} \quad t \geq 2 \end{cases} \tag{20}$$

and $(c_1, c_2) \in \mathcal{I}$, where

$$\mathcal{I} = \left\{(a,b) \in \{-6, -4, -2, 0, 2, 4, 6\}^2 \mid a \leq b\right\} \tag{21}$$

are enforced at the inlet $x = 0$.

**Geometric setup, boundary, and initial conditions.**   In the considered setup (see also Figure 3), a fluid flows past a fixed cylinder of radius $R = 0.05$ centered at $(x_c, y_c) = (0.2, 0.2)$ in a two-dimensional channel of length $L = 2.5$ and width $H = 0.41$. A deformable elastic strap of length $\ell = 0.35$ and height $h = 0.02$ is attached to the back of the cylinder. Note that, in the test cases considering fluid motion exclusively, the elastic strap is assumed to be rigid.

In the case of the fluid-structure interaction, the interaction conditions arise from the mechanical equilibrium at the boundaries of the strap, which are given by

$$\boldsymbol{\sigma}^f \cdot \mathbf{n} = \boldsymbol{\sigma}^s \cdot \mathbf{n}$$

$$u = \frac{\partial d}{\partial t}$$

where $\mathbf{n}$ denotes a unit normal vector to the fluid-solid interface. No-slip boundary conditions are imposed on the fluid velocity at the top (resp .bottom) boundaries of the channel at $y = 0$ (resp. $y = H$), as well as on the boundaries of the cylinder and the elastic strap. Outflow boundary conditions are imposed at $x = 2.5$ by enforcing the values $p = 0$ for the pressure.

The initial conditions

$$(u, p, d) = (0, 0, 0)$$

where the displacement $d = 0$ corresponds to a perfectly horizontal elastic strap and is imposed at time $t = 0$.

**Details regarding the data set generation.** The TurtleFSI package provides a monolithic solver for the fluid-structure interaction test case, that is, combining the equations describing the solid and fluid evolution into one coupled system based on an Arbitrary Eulerian-Lagrangian (ALE) formulation of the problem and developed on the FEniCS computing environment [44].

The initial conditions are expressed at set $X = X_{\mathcal{S}} \cup X_{\mathcal{F}}$ of mesh points, corresponding to the union of the solid and fluid domains. In the ALE formulation, at each snapshot $0 \leq t \leq t_M$ of the simulation, the solution is given at a set of mesh points $X_t = X + d_t$, where $d_t$ denotes the mesh displacement. In particular, the snapshots $u_t$ (resp. $p_t$) correspond to numerical approximations of the velocity (resp. the pressure) at the mesh points $X_t$. Notably, while equation (10) governs the deformation field in the solid domain $\Omega_t^s$, the displacements $d_t$ are obtained through an extension of the deformation field to the fluid domain $\Omega_t^f$ via a biharmonic extrapolation.

In all the cases considered, the values $\rho^f = 1.0 \times 10^3$, $\rho^s = 1.0 \times 10^3$, $\lambda^s = 4.0 \times 10^6$ and $\mu^s = 2.0 \times 10^6$ were used. The simulations were performed using a constant time step $\delta t = 0.01$.

## C.2 Justification of Experiment Design

For our setup, the fluid considered is water, with a density of 1000 kg.m-3 and a maximum inlet velocity of approximately $4 m.s^{-1}$, leading to Reynolds numbers in the range $200 - 2000$ ( $\mu = 10 - 1$) for our experiments. Only when the flow becomes turbulent can ample movements of the elastic strap (Fig. 4) be observed in the fluid-structure interaction case. Modeling fluid-solid interaction or only fluid motion with such a Reynolds number is quite challenging and used as a benchmark problem [17].

Modeling fluid-solid interaction with an even higher Reynolds number requires a very high computational cost. Because TurtleFSI's (used in this study) fluid solver, including its' fluid-structure interaction solver, uses a direct numerical simulation (DNS) of fluid dynamics and does not employ any turbulence models. This means that in order to accurately capture the small-scale energy-dissipating vortices that form when the flow interacts with the cylinder and strap at high Reynolds numbers, a very fine spatial domain discretization is required. Furthermore, an extremely small time step ($\Delta t$) is necessary to ensure numerical stability. For these reasons, the contribution [17], which introduced the benchmark fluid-structure interaction problem studied here, only deals with flows that have Reynolds numbers less than or equal to 200.

It's crucial to highlight a significant disparity between the pre-training and finetuning stages, particularly concerning examples with viscosities 1 and 10. This disparity arises from the utilization of distinct inlet boundary conditions during the pre-training and finetuning phases. Consequently, even though the viscosities align with the pre-training dataset during finetuning on PDEs featuring $\mu \in \{1, 10\}$, the model faces formidable challenges in adapting due to variations in inlet conditions. The finetuning dataset with viscosity=5 has different viscosity as well as intel conditions compared to the pre-training dataset, serving as an out-of-distribution PDE setup.

## C.3 Generating Rayleigh-Bénard dataset

The initial temperature field is initialized with a linear gradient between the hot bottom boundary, $\mathbf{T}_{\text{bottom}} = 1$, and the cold top boundary, $\mathbf{T}_{\text{top}} = 0$. To induce instability and initiate convection, temperature perturbations are introduced in localized regions of the domain. A region centered at $\left( \frac{L_x}{4}, \frac{L_y}{4} \right)$ is perturbed to $\mathbf{T} = 1$, while a region near the middle of the domain, centered at $\left( \frac{L_x}{2}, \frac{L_y}{2} \right)$,

is set to $\mathbf{T} = -1$. These perturbations break the symmetry and help to trigger the onset of convection patterns.

For the incompressible Navier-Stocks equation, we consider two-dimensional Kolmogorov flow (a form of the Navier-Stokes equations) for a viscous, incompressible fluid,

$$\frac{\partial \mathbf{u}}{\partial t} = -\mathbf{u} \cdot \nabla \mathbf{u} - \nabla p + \frac{1}{Re} \Delta \mathbf{u} + \sin(ny)\hat{\mathbf{x}}, \tag{22}$$

with the incompressibility constraint $\nabla \cdot \mathbf{u} = 0$ on the domain $[0, 2\pi]^2 \times (0, \infty)$. The initial condition is given as $\mathbf{u}(\cdot, 0) = \mathbf{u}_0$, where $\mathbf{u}$ denotes the velocity, $p$ the pressure, and $Re$ is the Reynolds number which we set to $500$ for our simulation.

## D  Comparison with FNO

We would like to bring out a distinction. In FNO, the mixing of channels happens in the Fourier space in the spectral layer through the linear transform $R$ applied to the Fourier coefficients. This defines a weighting of some sort on the input channels and how they are mixed. The mixing is global since the Fourier transform is a global operation. In CoDA-NO, however, the mixing of channels happens in the spatial domain through the attention mechanism as well as in the Fourier space because $\mathcal{K}^h, \mathcal{Q}^h, \mathcal{V}^h, \mathcal{M}$, and $\mathcal{I}$ are all implemented as FNO's. The attention weights determine how the channels are mixed, allowing for a more flexible and input-dependent mixing. This added flexibility enables CoDA-NO to better capture complex interactions and dependencies between different physical variables, especially in multiphysics problems where the relationships between variables can be intricate and vary depending on the input conditions. This also implies that this is more efficient as it can seamlessly incorporate additional or fewer variables during fine-tuning, avoiding retraining the whole model from scratch like FNO would have to, which can be computationally expensive.

## E  Visualization of Results

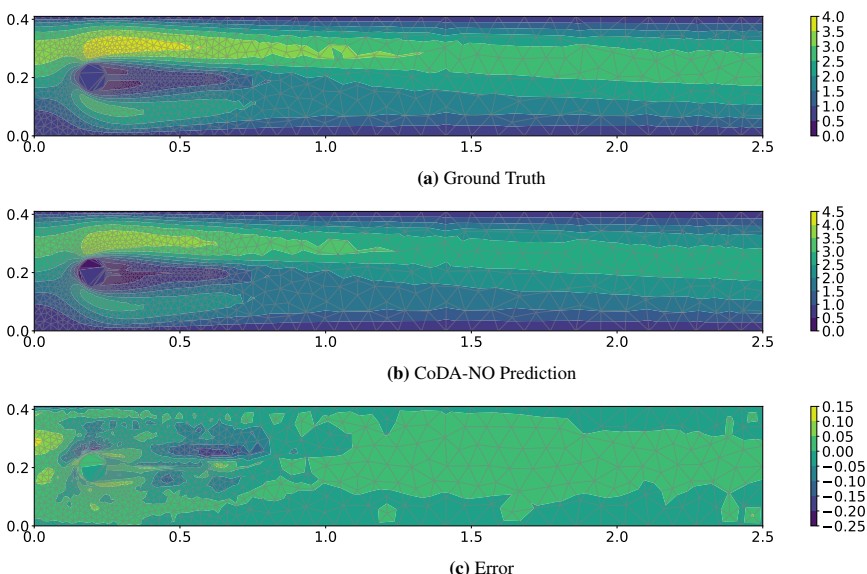

**(a)** Ground Truth

**(b)** CoDA-NO Prediction

**(c)** Error

**Figure 4.** Visualization of CoDA-NO prediction. We plot the horizontal velocity $u_x$ for the fluid-structure interaction problem.

# F  Additional Results

## F.1  Ablation of Proposed components

Table 3 shows that replacing the codomain attention with a regular attention mechanism or the removal of any of these designed components significantly impacts the model's performance. We also observe that our proposed normalization technique is crucial for effective training.

**Table 3. Evaluating $L_2$ loss across different models using various pre-training datasets and varying numbers of few-shot training samples.** "*" indicates configurations that did not converge due to excessive training error.

| CoDA-NO | VSPE | Norm | Pretrain Dataset | # Few Shot Training Samples | | | | | |
| --- | --- | --- | --- | --- | --- | --- | --- | --- | --- |
| | | | | 5 | | 25 | | 100 | |
| | | | | NS | NS+EW | NS | NS+EW | NS | NS+EW |
| ✗ | ✗ | ✗ | ✗ | 0.271 | 0.211 | 0.061 | 0.113 | 0.017 | 0.020 |
| ✓ | ✗ | ✗ | ✗ | 0.182 | 0.051 | 0.008 | 0.084 | 0.006 | 0.004 |
| ✓ | ✗ | ✓ | NS | 0.049 | 0.079 | 0.009 | 0.0132 | 0.004 | 0.009 |
| ✓ | ✗ | ✓ | NS EW | 0.045 | 0.057 | 0.010 | 0.011 | 0.008 | 0.004 |
| ✓ | ✓ | ✗ | NS | * | * | 0.023 | * | 0.008 | 0.006 |
| ✓ | ✓ | ✗ | NS EW | 0.057 | 0.232 | 0.012 | 0.052 | 0.006 | 0.006 |
| ✓ | ✓ | ✓ | NS | 0.025 | 0.071 | 0.007 | 0.008 | **0.004** | 0.005 |
| ✓ | ✓ | ✓ | NS EW | **0.024** | **0.040** | **0.006** | **0.005** | 0.005 | **0.003** |

## F.2  Zero-Shot Super Resolution Test

Here, we present the results of our zero-shot super-resolution models (see Tab. 4) on complex fluid-solid interaction problems. We train the models with 1317 mesh points on the domain (xy plain). However, during inference, the solution function is queried directly on a denser and non-uniform target mesh consisting of 2193 points.

We observe that the zero-shot super-resolution performance of CoDA-No is significantly better than the other baselines.

**Table 4. Zero Shot Super Resolution Performance on Fluid-Solid (NS-EW) Interaction Problem**

| Model | Pretrain Dataset | Fluid Viscosities | | |
| --- | --- | --- | --- | --- |
| | | $\mu = 5$ | $\mu = 1$ | $\mu = 10$ |
| U-Net | - | 0.144 | 0.267 | 0.216 |
| Vit | - | 0.052 | 0.175 | 0.046 |
| GINO | - | 0.069 | 0.103 | 0.0711 |
| DeepO | - | 0.113 | 0.107 | 0.357 |
| GNN | - | 0.223 | 0.211 | 0.247 |
| CoDA-NO | NS-ES | 0.041 | 0.063 | 0.048 |
| CoDA-NO | NS | **0.032** | **0.049** | **0.035** |

## F.3  Parameter Count and Computational Cost.

Now the present the number of parameters and training/interference time taken by the proposed model along with different baselines used in the study in Tab. 5. It might seem that models are not compared fairly, as the CoDA-NO has a higher parameter count. However, here, we test the models on a few shot learning problems. Increasing the baselines' parameter count worsens the overfitting problem.

To demonstrate this fact, we perform experiments on a fluid-solid interaction dataset with an increased parameter count. We will observe that increasing the parameter count almost always negatively impacts the performance, especially for very few hot learning scenarios (see Tab. 6).

We also note that the additional model parameters and computation are required to learn rich inter-variable dependencies during pre-training and generalize from single to multi-physics during finetuning. Furthermore, the zero-shot super-resolution capability of CoDA-NO is discussed in Sec. F.2. CoDA-NO is a justified choice due to its seamless adaptation to various PDEs, remarkable performance gap, and zero-shot super-resolution capability despite having a little more computational overhead.

**Table 5. Comparison of Inference Time, Training Time (in sec.) per sample, and Number of Parameters for different models.**

| Models | GNN | GINO | DeepO | ViT | Unet | CoDA-NO |
|---|---|---|---|---|---|---|
| Inference Time | 0.012 | 0.012 | 0.006 | 0.071 | 0.024 | 0.440 |
| Training Time | 0.136 | 0.136 | 0.131 | 0.273 | 0.268 | 1.250 |
| # Parameter $\times 10^6$ | 0.6 | 60 | 6 | 27 | 30 | 43 |

**Table 6. Overfitting of Baselines with Higher Parameters (in $\times 1e6$) on NS-EW dataset**

| Models | # Parameter (Used/High) | # Train = 5 (Used / High) | # Train=25 (Used/ High) | # Train=100 (Used / High) |
|---|---|---|---|---|
| GINO | 60/200 | 0.122 / 0.342 | 0.053 / 0.066 | 0.043 / 0.036 |
| DeepO | 6 / 25 | 0.482 / 0.495 | 0.198 / 0.303 | 0.107 / 0.083 |
| GNN | 0.6/7 | 0.045 / 0.268 | 0.009 / 0.031 | 0.009 / 0.061 |
| ViT | 27/100 | 0.211 / 0.266 | 0.113 / 0.125 | 0.020 / 0.022 |
| U-net | 30/48 | 3.579 / 9.462 | 0.842 / 3.957 | 0.203 / 0.412 |

### F.4 Energy Spectrum

Here, we show the energy spectrum for the NS-EW dataset for $\mu = 5$ calculated from the test set (see Fig. 5). All models are trained on 100 training examples. Due to numerical error, the measured spectral energy does not decay smoothly in the high-frequency region. However, our models' energy spectrum remains closest to the ground truth.

### F.5 Ablation on Finetuning Technique

Here, in Tab. 7, Tab. 9, and Tab. 8, we present some additional ablation studies on our model's performance when we keeping the weight of the "Encoder" frozen during supervised fine-tuning.

### F.6 Error bar

## G PDEBench experiments

We finally compare CoDA-NO, FNO, and the recently proposed DPOT [9] on PDEBench [19]. DPOT (Auto-Regressive Denoising Operator Transformer for Large-Scale PDE Pre-Training) is a large-scale pre-training approach for learning PDE representations. It utilizes an auto-regressive denoising strategy and a Fourier transformer architecture for efficient pre-training on diverse PDE datasets.

To assess the performance of these models on a diverse set of PDEs, we conduct experiments on three single-physics datasets from the PDEBench: Shallow Water Equations (SWE), Diffusion Equations (DIFF), and Navier-Stokes Equations (NS). We follow the same pretraining and finetuning procedure for both models' datasets. And we evaluate the models on the following two tasks

- **Reconstructive task**: The models are trained on the respective datasets during pretraining to use the self-supervised learning objective to reconstruct the masked input. The error

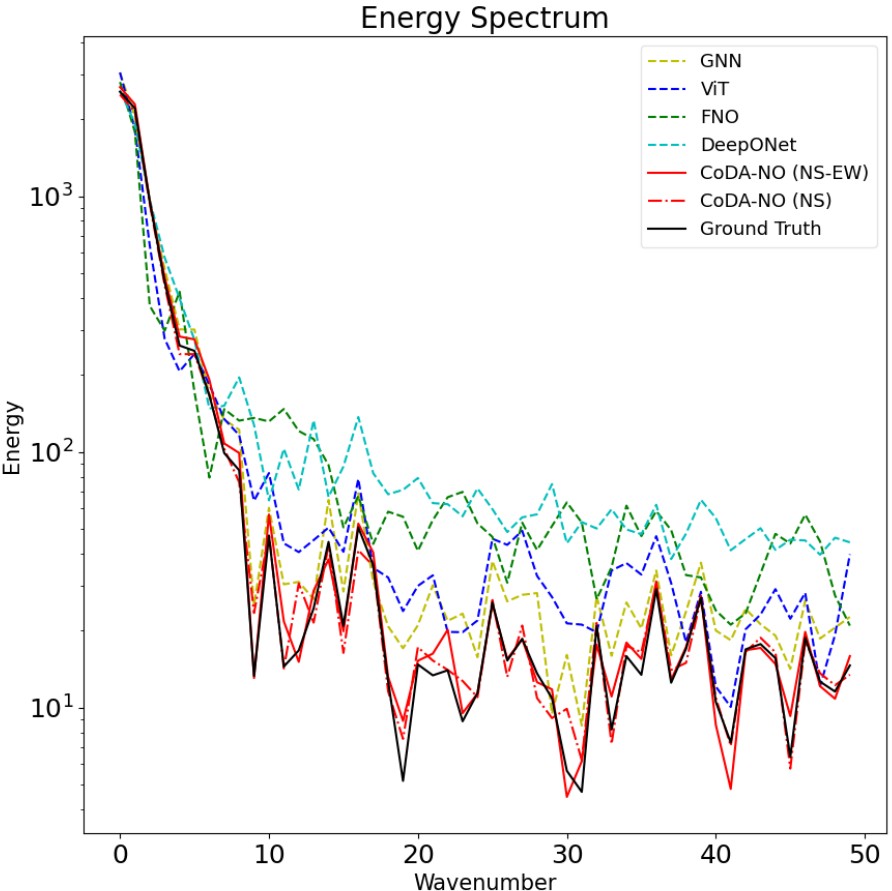

**Figure 5.** Energy Spectrum of the Velocity Field of the fluid on the fluid-solid interaction dataset.

**Table 7. Test errors ($L_2$ loss) for fluid dynamics (NS) and fluid-solid interaction (NS+EW) datasets with viscosity** $\mu = 1.0$ **for different numbers of few-shot training samples.** The pre-training is done with 8000 samples taken from NS and NS+EW datasets with viscosities $\mu \in \{1.0, 10.0\}$.

| Model | Pretrain Dataset | # Few Shot Training Samples | | | | | | | | | |
|---|---|---|---|---|---|---|---|---|---|---|---|
| | | 5 | | 10 | | 50 | | 100 | | 250 | |
| | | NS | NS+EW | NS | NS+EW | NS | NS+EW | NS | NS+EW | NS | NS+EW |
| Ours | NS | 0.0493 | 0.2645 | 0.0237 | 0.1955 | 0.0092 | 0.0378 | 0.0103 | 0.0604 | 0.0085 | 0.0294 |
| | NS+EW | 0.0416 | 0.2371 | 0.0221 | 0.1786 | 0.0105 | 0.0484 | 0.0110 | 0.0380 | 0.0089 | 0.0273 |
| CoDA-NO | - | 0.1279 | 0.2435 | 0.0225 | 0.2282 | 0.0117 | 0.0745 | 0.0115 | 0.0219 | 0.0091 | 0.0148 |
| GINO | - | 0.3337 | 0.2615 | 0.3189 | 0.1817 | 0.0596 | 0.0667 | 0.0349 | 0.0636 | 0.0209 | 0.0308 |
| GNN | - | 0.0265 | 0.1800 | 0.0222 | 0.1799 | 0.0068 | 0.0867 | 0.0113 | 0.0539 | 0.0050 | 0.0193 |
| ViT | - | 0.2738 | 0.5087 | 0.1519 | 0.4146 | 0.0473 | 0.1119 | 0.0407 | 0.1106 | 0.0119 | 0.0381 |
| U-Net | - | 25.33 | 1.434 | 4.007 | 4.320 | 0.1495 | 0.6653 | 0.07723 | 0.1821 | 0.0934 | 0.1651 |
| DeepONet | - | 1.262 | 0.8186 | 0.6485 | 0.4937 | 0.2576 | 0.3198 | 0.1992 | 0.3399 | 0.1385 | 0.1916 |

achieved through this is called the ***"Reconstruction error"***. This allows the models to learn meaningful representations of the underlying physical systems.

Table 8. Test errors ($L_2$ loss) for fluid dynamics (NS) and fluid-solid interaction (NS+EW) datasets with viscosity $\mu = 5.0$ **for different numbers of few-shot training samples.** The pre-training is done with 8000 samples taken from NS and NS+EW datasets with viscosities $\mu \in \{1.0, 10.0\}$.

| Model | Pretrain Dataset | # Few Shot Training Samples | | | | | | | | | |
| | | 5 | | 10 | | 50 | | 100 | | 250 | |
| | | NS | NS+EW | NS | NS+EW | NS | NS+EW | NS | NS+EW | NS | NS+EW |
|---|---|---|---|---|---|---|---|---|---|---|---|
| Ours | NS | 0.0190 | 0.1597 | 0.0141 | 0.0220 | 0.0043 | 0.0042 | 0.0054 | 0.0053 | 0.0033 | 0.0025 |
| | NS+EW | 0.0201 | 0.1077 | 0.0157 | 0.0153 | 0.0053 | 0.0053 | 0.0044 | 0.0030 | 0.0037 | 0.0022 |
| CoDA-NO | - | 0.1820 | 0.0513 | 0.0107 | 0.0199 | 0.0063 | 0.0066 | 0.0062 | 0.0045 | 0.0041 | 0.0029 |
| GINO | - | 0.2004 | 0.1222 | 0.2245 | 0.0753 | 0.0359 | 0.0364 | 0.0222 | 0.0438 | 0.0163 | 0.0190 |
| GNN | - | 0.0390 | 0.0460 | 0.0280 | 0.0294 | 0.0045 | 0.0123 | 0.0086 | 0.0094 | 0.0064 | 0.0033 |
| ViT | - | 0.2719 | 0.2113 | 0.1889 | 0.1561 | 0.0271 | 0.0474 | 0.0173 | 0.0207 | 0.0077 | 0.0122 |
| U-Net | - | 13.3370 | 3.5790 | 1.1540 | 2.1340 | 0.1608 | 0.3178 | 0.1418 | 0.2035 | 0.1317 | 0.1180 |
| DeepONet | - | 0.6863 | 0.4821 | 0.6720 | 0.2945 | 0.2019 | 0.2024 | 0.1076 | 0.1070 | 0.0731 | 0.1085 |

Table 9. Test errors ($L_2$ loss) for fluid dynamics (NS) and fluid-solid interaction (NS+EW) datasets with viscosity $\mu = 10.0$ **for different numbers of few-shot training samples.** The pre-training is done with 8000 samples taken from NS and NS+EW datasets with viscosities $\mu \in \{1.0, 10.0\}$.

| Model | Pretrain Dataset | # Few Shot Training Samples | | | | | | | | | |
| | | 5 | | 10 | | 50 | | 100 | | 250 | |
| | | NS | NS+EW | NS | NS+EW | NS | NS+EW | NS | NS+EW | NS | NS+EW |
|---|---|---|---|---|---|---|---|---|---|---|---|
| Ours | NS | 0.0186 | 0.1203 | 0.0105 | 0.0207 | 0.00327 | 0.00444 | 0.00391 | 0.00412 | 0.00229 | 0.00215 |
| | NS+EW | 0.0171 | 0.0925 | 0.0109 | 0.0130 | 0.00383 | 0.00360 | 0.00303 | 0.00232 | 0.00225 | 0.00133 |
| CoDA-NO | - | 0.0859 | 0.0618 | 0.0115 | 0.0166 | 0.00494 | 0.00763 | 0.00660 | 0.00330 | 0.00374 | 0.00195 |
| GINO | - | 0.2316 | 0.1560 | 0.1679 | 0.0582 | 0.04122 | 0.03327 | 0.03074 | 0.0395 | 0.01389 | 0.01139 |
| GNN | - | 0.0715 | 0.0448 | 0.0547 | 0.0179 | 0.00789 | 0.00494 | 0.00319 | 0.0148 | 0.00547 | 0.00229 |
| ViT | - | 0.4001 | 0.2201 | 0.3388 | 0.1967 | 0.06215 | 0.05700 | 0.04299 | 0.01867 | 0.00770 | 0.00903 |
| U-Net | - | 1.2550 | 0.6995 | 0.3255 | 0.9148 | 0.3156 | 0.2095 | 0.1889 | 0.2085 | 0.1732 | 0.3132 |
| DeepONet | - | 0.7158 | 0.3794 | 0.5515 | 0.2533 | 0.1164 | 0.1337 | 0.1027 | 0.07814 | 0.08359 | 0.05697 |

Table 10. Error bar representing standard deviation over three runs with different number of few shot example for NS+EW dataset for $Re = 400$

| Models | Pre-training Dataset | # Few Shot Training Samples | | |
| | | 5 | 25 | 100 |
|---|---|---|---|---|
| GINO | | $0.121 \pm 0.023$ | $0.0530 \pm 0.0053$ | $0.0345 \pm 0.0086$ |
| DeepO | | $0.534 \pm 0.005$ | $0.1920 \pm 0.0072$ | $0.1384 \pm 0.0293$ |
| GNN | | $0.121 \pm 0.136$ | $0.0304 \pm 0.0210$ | $0.0200 \pm 0.0120$ |
| ViT | | $0.276 \pm 0.093$ | $0.0837 \pm 0.0284$ | $0.0208 \pm 0.0044$ |
| U-net | | $1.770 \pm 1.636$ | $0.8368 \pm 0.3503$ | $0.5814 \pm 0.5680$ |
| Ours | | $0.059 \pm 0.017$ | $0.0096 \pm 0.0010$ | $0.0038 \pm 0.0003$ |
| | NS | $0.068 \pm 0.055$ | $0.0078 \pm 0.0002$ | $0.0036 \pm 0.0005$ |
| | NS-EW | $0.044 \pm 0.041$ | $0.0057 \pm 0.0012$ | $0.0034 \pm 0.0006$ |

- **Predictive task**: Subsequently, we finetune the pre-trained models using a supervised learning objective, where the goal is to minimize the *"Prediction error"* by accurately predicting the next 5 timesteps given the input history. We try to learn the solution operator that maps the state of the system from time $t \in [0, T]$ to the state at time $t \in [T, T + 5]$, effectively predicting the next 5 timesteps given the history up to time $T$.

Both models were pre-trained and fine-tuned on this dataset for 35 epochs.

**Table 11.** Error bar representing standard deviation over three runs with different number of few shot example for NS dataset for $Re = 400$

| Models | Pre-training Dataset | # Few Shot Training Samples | | |
|---|---|---|---|---|
| | | 5 | 25 | 100 |
| GINO | | $0.143 \pm 0.0231$ | $0.0371 \pm 0.0044$ | $0.0330 \pm 0.0105$ |
| DeepO | | $0.621 \pm 0.2417$ | $0.3162 \pm 0.1146$ | $0.1978 \pm 0.0345$ |
| GNN | | $0.021 \pm 0.0124$ | $0.0046 \pm 0.0012$ | $0.0051 \pm 0.0024$ |
| ViT | | $0.196 \pm 0.0326$ | $0.0409 \pm 0.0057$ | $0.0302 \pm 0.0160$ |
| U-net | | $7.241 \pm 4.3200$ | $0.6568 \pm 0.3635$ | $0.2025 \pm 0.1223$ |
| | | $0.0612 \pm 0.0364$ | $0.0094 \pm 0.0006$ | $0.0045 \pm 0.0006$ |
| Ours | NS | $0.0276 \pm 0.0032$ | $0.0057 \pm 0.0005$ | $0.0039 \pm 0.0001$ |
| | NS-EW | $0.0273 \pm 0.0054$ | $0.0056 \pm 0.0005$ | $0.0040 \pm 0.0001$ |

**Table 12.** Test errors($L_2$ error) for CoDA-NO vs FNO on 2D datasets from PDEBench. SWE indicates shallow water equations, DIFF indicates the diffusion equation. NS+DIFF+SWE means pretraining and fine-tuning on a combined Navier-Stokes, diffusion, and shallow water equations dataset.

| Model | Dataset | Test Error | |
|---|---|---|---|
| | | Prediction Error | Reconstruction Error |
| CoDA-NO | SWE | **0.04072** | **0.00460** |
| FNO | | 0.04631 | 0.03262 |
| CoDA-NO | DIFF | **0.00810** | **0.00041** |
| FNO | | 0.01415 | 0.01894 |
| CoDA-NO | NS+DIFF+SWE | 0.00302 | **0.00006** |
| FNO | | **0.00118** | 0.00287 |

Table 12 presents CoDA-NO and FNO [12] test errors on the single-physics PDEs datasets sourced from PDEBench [19]. For the single-physics experiments, CoDA-NO consistently outperforms FNO, improving generalization (predictive error) up to **43%** over FNO, indicating its ability to capture complex dynamics and dependencies within these systems. We report additional details and experimental results in Sec. G comparing FNO, DPOT [9] and CoDA-NO where CoDA-NO demonstrates superior performance and parameter efficiency. It is worth noting that DPOT and MPP [8] are significantly bigger models but can also handle a larger set of PDEs.

In addition to the single-physics experiments, we also explore the potential of joint pretraining and finetuning across multiple PDE systems. We create a combined dataset by merging the SWE, DIFF, and NS datasets, even though these PDEs do not share any common physical variables or governing equations. Both FNO and CoDA-NO were pre-trained and fine-tuned on this dataset for 35 epochs.

DPOT, with its large-scale pre-training approach, demonstrates strong performance on the SWE and DIFF datasets. Even with a 500 million parameter model (DPOT-L-500), DPOT achieves impressive results, reducing the test error to 0.0017 on SWE and 0.0073 on DIFF. It is also important to note that DPOT is larger as it was pretrained on 12 different datasets, and hence the size is justified. However, it is noteworthy that CoDA-NO, with only 11 million parameters, comes very close to achieving similar generalization performance. CoDA-NO's test error on DIFF (0.0081) is comparable to DPOT's performance, as shown in table 15, despite having significantly fewer parameters and fewer finetuning epochs. However, on the other hand, we see that CoDA-NO doesn't do well on the SWE dataset as shown in table 14; we assume that the case would be the fact that we would need to finetune for more epochs to achieve better results. The SWE task is also a harder dataset; increasing the model complexity and pretraining epochs would help get better results.

It is important to highlight that DPOT was pre-trained on 12 datasets for 1000 epochs, while CoDA-NO was pre-trained and fine-tuned on a single dataset for 35 epochs. Despite this difference in pre-training data and epochs, CoDA-NO still achieves competitive results compared to DPOT's 200/500 epochs of fine-tuning.

These results suggest that when there is shared physics between the pre-training and fine-tuning datasets, CoDA-NO can effectively leverage this commonality to achieve strong generalization

**Table 13. Comparison of model parameter sizes for CoDA-NO, FNO, and DPOT.** DPOT-FT stands for the Finetuning model used, whereas -T stands for tiny, -S stands for small, -M stands for medium, and -L stands for Large. The pretrained model sizes are present in the original paper but are around the same parameter sizes as the fine-tuned models.

| Model | Model Parameters |
|---|---|
| CoDA-NO | 11M |
| FNO | 1.9B |
| DPOT-FT-T | 7M |
| DPOT-FT-S | 30M |
| DPOT-FT-M | 100M |
| DPOT-FT-L | 500M |

**Table 14. Test errors for CoDA-NO vs DPOT on 2D datasets from PDEBench. SWE indicates shallow water equations data.** '12DATA' represents the 12PDE PDE datasets DPOT is pretrained on. The "-200" and "-500" suffixes denote fine-tuning on each subset for 200 and 500 epochs, respectively, which is directly taken from the DPOT paper. All of this was fine-tuned on SWE data.

| Model | Pretrained Dataset | Predcition Error |
|---|---|---|
| CoDA-NO | SWE | 0.0407 |
| FNO | SWE | 0.0463 |
| T-200 | 12DATA | 0.0028 |
| S-200 | 12DATA | 0.0022 |
| M-200 | 12DATA | 0.0021 |
| L-200 | 12DATA | 0.0019 |
| T-500 | 12DATA | 0.0024 |
| S-500 | 12DATA | 0.0023 |
| M-500 | 12DATA | 0.0022 |
| L-500 | 12DATA | **0.0017** |

performance. However, when there is no shared physics, as in the case of the combined dataset, CoDA-NO's performance may not be as remarkable.

Table 13 compares the model sizes of CoDA-NO, FNO, and DPOT. CoDA-NO's model size of 11 million parameters is significantly smaller than FNO's 1.9 billion parameters and DPOT's largest model size of 500 million parameters. This highlights CoDA-NO's parameter efficiency and its ability to achieve competitive performance with a more compact model.

In summary, these experiments on the PDEBench datasets demonstrate the effectiveness of CoDA-NO in learning and generalizing to different PDE systems. CoDA-NO's performance, especially considering its smaller model size and shorter pre-training, showcases its potential as a foundation model for scientific machine learning. The ability to achieve competitive results with DPOT, despite the differences in pre-training data and epochs, further highlights CoDA-NO's efficiency and generalization capabilities.

### G.1 Ablation on the Size of FNO

In Tab. 16, we present the performance of FNO with a different number of parameters along with the performance of CoDA-NO.

## H Implementation Details

The variable encoders are implemented using a multi-layer perceptron mapping the position $x \in \mathcal{D}$ to the embedding vector. Following transformers and NeRF [45] model, we use positional encoding

**Table 15.** Test errors for CoDA-NO vs DPOT on 2D datasets from PDEBench. DIFF indicates the diffusion equation data. '12DATA' represents the 12PDE datasets DPOT was trained on. The "-200" and "-500" suffixes denote fine-tuning on each subset for 200 and 500 epochs, respectively, which is directly taken from the DPOT paper. All of this was fine-tuned on DIFF data.

| Model | Pretrained Dataset | Test error |
|---|---|---|
| CoDA-NO | DIFF | **0.0081** |
| FNO | DIFF | 0.0141 |
| T-200 | 12DATA | 0.0194 |
| S-200 | 12DATA | 0.0171 |
| M-200 | 12DATA | 0.0142 |
| L-200 | 12DATA | 0.0158 |
| T-500 | 12DATA | 0.0148 |
| S-500 | 12DATA | 0.0129 |
| M-500 | 12DATA | 0.0103 |
| L-500 | 12DATA | **0.0073** |

**Table 16.** Error in $L_2$ norm for models in both the Shallow Water Equation and Diffusion-Reaction experiments. The number of parameters is reported alongside the $L_2$ errors for both tasks.

| Model | # Parameter | $L_2$ (SWE) | $L_2$ (DIFF) |
|---|---|---|---|
| CoDA-NO | 11M | 0.0407 | 0.0081 |
| FNO | 1.9B | 0.0463 | 0.0141 |
| FNO | 485M | 0.0424 | 0.0145 |
| FNO | 120M | 0.0410 | 0.0153 |
| FNO | 11M | 0.0491 | 0.0268 |
| FNO | 1M | 0.2355 | 0.2085 |

instead of raw coordinates. The Encoder and Reconstructor modules use three stacked CoDA-NO layers. The Predictor modules use one layer of CoDA-NO.

For every training sample, one of the following two masking choices is selected with equal probability

- 50% of the mesh points of 60% of variables are masked.
- 30% of the variables are masked out completely.

In order to apply masking on an irregular mesh, we select a point at random from the mesh. Following this, we identify the neighboring points within a fixed distance from the selected point and set their values to zero. This process is continued until we have masked out a predetermined portion of all mesh points.

