# OpenReview forum: "Pretraining Codomain Attention Neural Operators for Solving Multiphysics PDEs"
_NeurIPS.cc/2024/Conference — NeurIPS 2024 poster_

### Official Review · Reviewer_ZkA5 · 2024-07-01

**Soundness:** 3
**Presentation:** 3
**Contribution:** 2
**Rating:** 5
**Confidence:** 4

**Summary:**

The authors introduced an innovative attention-based neural operator and evaluated it against various baselines. They employed masked pretraining and finetuning techniques, comparing the model's performance to multiple benchmarks. Their study included interesting problems such as fluid-structure interactions. The authors showed that their approach is effective for few-shot learning in their experimental evaluations.

**Strengths:**

- The paper presents a new neural operator architecture based on attention mechanisms. This architecture demonstrates superior performance compared to tested baseline models on the NS and NS+EW benchmarks, highlighting its potential advancements in solving PDE-related problems.
- To the best of my knowledge, the authors are the first ones to use masked training for PDE learning effectively
- The mathematical formulation of the proposed model is well-articulated in the paper. This clarity helps readers understand the underlying principles of the model's operation.
- The study addresses a compelling and very relevant multiphysics problem involving fluid-structure interactions.
- The authors demonstrated through empirical evidence that their approach is effective for few-shot finetuning in various scenarios.

**Weaknesses:**

- The study in Table 1 demonstrates the model’s performance on two specific PDEs: Navier-Stokes for fluid flow and a coupled Navier-Stokes with elastodynamics equations for fluid-solid interactions. While these cases provide some insight into the model's capabilities, they are not sufficient to generalize the model's applicability to a broader range of multiphysics problems.
- For the NS dataset with Reynolds number Re=400, the model trained from scratch with only 25 samples matches the performance of the pretrained model. In the case of NS+EW benchmark, when the Reynolds number increases to 4000, even with just 5 samples, both the finetuned and scratch-trained models exhibit similar testing errors. This suggests that pretraining may not provide significant advantages in many cases.
- The use of the L2 loss metric to evaluate model performance is problematic because it aggregates outputs of different physical meanings, such as pressure p, velocity u, and displacement d, into a single loss value. This can obscure individual variable contributions and lead to misleading conclusions about model accuracy.
- The absence of prediction visualizations diminishes the interpretability of the L2 loss values. Visualizing predictions could provide more intuitive insights into model performance and clarify discrepancies in the loss metric.
- The study in the Table 1 does not include a Fourier Neural Operator. Including such a benchmark is crucial to fairly evaluate CoDA-NO’s performance against an FNO model of similar size.
- The FNO model used for comparison in Table 2 has 1.9 billion parameters, vastly outnumbering the CoDA-NO's 11 million parameters. This overparameterization likely affects the model's performance due to the relatively small training set sizes and makes the comparison with CoDA-NO’s performance misleading. Smaller FNO models could provide a more realistic performance benchmark. The claim that CoDA-NO’s better performance compared to a much larger FNO model demonstrates parameter efficiency is misleading. Parameter efficiency should be evaluated with models of comparable sizes, and overparameterized models may not reflect typical scenarios.
- CoDA-NO has significantly higher inference times compared to other baseline models.

These points collectively highlight the need for more comprehensive experiments, appropriate metrics, realistic model comparisons, and practical considerations like inference time to fully evaluate the model's capabilities.

**Questions:**

- What is the reasoning behind applying the attention in the in the channel space? Do the models scale better? Do they have higher expresive power?
- Why is 1.9B FNO model used for comparisson?

**Limitations:**

The authors explained the limitations.

---

> ### Author Rebuttal · Authors · 2024-08-07
>
> We appreciate that the reviewer values our work and recognizes its novelty and the mathematical formulation of the proposed model.
> ```
> Q1. Evaluation on additional PDE datasets
> ```
> In addition to the coupled fluid-solid interaction problem, we also provide experiments on different PDE systems (please see Table 2 and Appendix Sec E, Table 11-12), where we test the proposed CoDA-NO architecture on shallow water equation, diffusion equation, and Navier Stokes equation.
> In addition, we also include the results of our study on the Rayleigh-Benard convection system. The CoDA-NO is pre-trained in a supervised way on the Navier–Stokes (NS) equations. Then it is subsequently finetuned on a limited few shot data ($N \in \{5,10,25\}$) of Rayleigh-Benard convection. We show the result in the following table, where we report the $L2$ loss. We see that, in the extremely low data regime, pre-trained CoDA-NO performs better than FNO and CoDA-NO trained from scratch.
> | Model    | Pre-training dataset | N=5      | N=10    | N=25    |
> | -------- | --------                           | -------- |-----        |-----        |
> | FNO      | -                                | 0.439    |0.249    |0.130    |
> | CoDA-NO  | -                           | 0.315    |0.272    |0.203    |
> | CoDA-NO  | NS                      | 0.2537   |0.2223   |0.179    |
>
> Please note that these results are preliminary. As per reviewer qRRk's suggestion, we are conducting a comprehensive study on the Rayleigh-Benard convection system.
>
> ```
> Q2. Performance gain from pre-training.
> ```
> In most cases, pre-trained CoDA-NO demonstrated a significant performance improvement compared to CoDA-NO trained from scratch. While we acknowledge that in some instances, the difference is not substantial, it is consistently better. Additionally, in general, pre-trained models offer benefits such as reusability and faster convergence with fewer epochs required. The lack of a significant performance gain in a few settings does not diminish the importance and potential advantages of adapting the architecture and pre-training scheme. Further, our self-supervised pretraining method is also beneficial when the cost of generating training data with numerical solvers is very high.
>
> ```
> Q3. Per variable loss
> ```
>
> Previous studies have shown that aggregated loss demonstrates model accuracy across multiple variables [1, 2, 3] which helps to get a comprehensive overview of the performamnce. For instance, the results of the compressible NS equation (velocity and pressure field) are presented in an aggregated manner [1, 2, 3].
> We also provide the per-variable performance of each of the variables (velocity $v_x, v_y$, pressure $p$, displacement $d_x,d_y$) on the combined Navier-Stokes and Elastic wave equation for $Re=4000$.
> ### Table of Horizontal velocity ($u_x$)
> | Model | Pretrain Dataset |  N=5  |  N=25 | N=100 |
> |-------|:----------------:|:-----: |:-----:|:-----:|
> | GINO  |         -        | 0.416 | 0.145 | 0.105 |
> | DeepO |         -       | 0.513 | 0.597 | 0.189 |
> | GNN   |         -        | 0.056 | 0.186 | 0.119 |
> | ViT   |         -          | 0.393 | 0.710 | 0.437 |
> | U-Net |         -        | 2.936 | 0.227 | 0.174 |
> | CoDA-NO  |         -        | 0.078 | 0.048 | 0.023 |
> |  CoDA-NO   |        NS    | 0.061 | 0.034 | 0.016 |
> |   CoDA-NO |NS+EW      | 0.047 | 0.020 | 0.010 |
>
> ### Table of Verticle velocity ($u_y$)
> | Models  | Pretrain  Dataset |  N=5  |  N=25 |  N=100 |
> |---------|:---------------------:|:-----:|:-----:|:------:|
> | GINO    |           -           | 0.503 | 0.403 |  0.111 |
> | DeepO   |           -           | 1.006 | 0.865 |  0.688 |
> | GNN     |           -           | 0.462 | 0.270 |  0.169 |
> | ViT     |           -           | 1.334 | 0.726 | 0.4254 |
> | U-Net   |           -           | 0.967 | 0.417 |  0.269 |
> | CoDA-NO |           -           | 0.240 | 0.153 | 0.0714 |
> | CoDA-NO |           NS          | 0.233 | 0.132 |  0.067 |
> | CoDA-NO |         NS+EW         | 0.208 | 0.120 |  0.060 |
>
> ### Table of Verticle velocity ($p$)
> | Models  | Pre-train dataset |  N =5 |  N =25 | N =100 |
> |---------|:-----------------:|:-----:|:------:|:------:|
> | GINO    |         -         | 0.478 |  0.152 |  0.110 |
> | DeepO   |         -         | 1.435 |  0.337 |  0.229 |
> | GNN     |         -         | 0.209 |  0.149 |  0.110 |
> | ViT     |         -         | 0.560 |  0.257 |  0.148 |
> | U-Net   |         -         | 0.581 |  0.337 |  0.229 |
> | CoDA-NO |         -         | 0.174 | 0.2251 |  0.077 |
> | CoDA-NO |         NS        | 0.526 | 0.1326 |  0.064 |
> | CoDA-NO |       NS+EW       | 0.398 |  0.097 | 0.0591 |
>
> ### Table of Horizontal displacement($d_x$)
> | Models  | Pre-training Dataset |  N=5  |  N=25 | NS+EW |
> |---------|:--------------------:|:-----:|:-----:|:-----:|
> | GINO    |           -          | 1.326 | 0.353 | 0.117 |
> | DeepO   |           -          | 1.467 | 0.354 | 0.216 |
> | GNN     |           -          | 0.654 | 0.466 | 0.240 |
> | ViT     |           -          | 0.769 | 0.474 | 0.279 |
> | U-Net   |           -          | 1.083 | 0.730 | 0.365 |
> | CoDA-NO |           -          | 0.582 | 0.670 | 0.139 |
> | CoDA-NO |          NS          | 0.631 | 0.372 | 0.156 |
> | CoDA-NO |         NS+EW        | 0.531 | 0.350 | 0.147 |

---

> ### Author Response · Authors · 2024-08-07
> **Rebuttal**
>
> ### Vertical displacement ($d_y$)
> | Models  | Pre-training dataset | NS+EW | NS+EW | NS+EW |
> |---------|:--------------------:|:-----:|:-----:|:-----:|
> | GINO    |           -          | 0.841 | 0.238 | 0.102 |
> | DeepO   |           -          | 0.760 | 0.171 | 0.050 |
> | GNN     |           -          | 0.561 | 0.322 | 0.162 |
> | ViT     |           -          | 1.223 | 0.335 | 0.068 |
> | U-Net   |           -          | 6.060 | 0.297 | 0.265 |
> | CoDA-NO |           -          | 0.576 | 0.331 | 0.061 |
> | CoDA-NO |          NS          | 0.381 | 0.138 | 0.085 |
> | CoDA-NO |         NS+EW        | 0.355 | 0.129 | 0.065 |
>
> [1] Dpot: Auto-regressive denoising operator transformer for large-scale PDE pre-training.
>
> [2] Gnot: A general neural operator transformer for operator learning.
>
> [3] Pdebench: An extensive benchmark for scientific machine learning.
>
>
> ```
> Q4. Visualization of output
> ```
> We provide visualization in the attached PDF (Figure 1). We will include more visualizations in the paper.
> ```
> Q5. FNO as a baseline for fluid-solid interaction problem
> ```
> As the domain's discretization is irregular, we don't directly apply FNO since the computationally feasible implementation of such models only supports uniform grids (through the deployment of FFT). However, for the shallow water equation, for which the discretization is uniform, we use the spherical variants of FNO as our baseline.
>
>
> ```
> Q6. Parameterization of FNO
> ```
> For the experiment on the PDEbench (Table 2, Table 11-12), we do a different setup compared to the fluid-solid interaction. In this case, we follow the same setup as DPOT and first pretrain the model using self-supervised learning and then fine tune with supervised loss, and in our setting, both are evenly split.  Due to this relatively large dataset for both pretraining and fine-tuning, the risk of overfitting is low.  Nevertheless, we have done an extensive ablation of the number of parameters of FNO, where the model is first trained in a self-supervised manner followed by supervised training. We report the $L_2$ for each of the datasets. We can see that FNO with 1.9B parameter does not overfit the data but rather performs better. We hypothesize that this is due to self-supervised pre-training done before supervised training and the size of the training set.
>
>
>
> ### Table for shallow water equation
> | Model    | # parameter | $L2$ |
> |----------|:-----------:|:-----------------:|
> | CoDA-NO  |     11M     |       0.0407      |
> | FNO      |     1.9B    |       0.0463      |
> | FNO      |     485M    |       0.0424      |
> | FNO      |     120M    |       0.0410      |
> | FNO      |     11M     |       0.0491      |
> | FNO      |      1M     |       0.2355      |
>
> ### Table for diffusion-reaction
> | Model    | # Parameter |  $L_2$ |
> |----------|:-----------:|:------:|
> | CoDA-NO  |     11M     | 0.0081 |
> | FNO      |     1.9B    | 0.0141 |
> | FNO      |     485M    | 0.0145 |
> | FNO      |     120M    | 0.0153 |
> | FNO      |     11M     | 0.0268 |
> | FNO      |      1M     | 0.2085 |
>
>
> ```
> Q7. Higher inference time
> ```
>  The objective of the proposed CoDA-NO is to provide suitable architecture for foundation models for solving PDEs and multi-physics problems. In this study, we demonstrated the extensive adaptability and advantages of CoDA-NO and its pre-training mechanism. Similar to other large foundation models (such as [1]), CoDA-NO has a higher inference time compared to a smaller model but is still significantly faster compared to numerical solvers. Furthermore, the computation time can be mitigated through careful implementation, parallel computation, and engineering efforts, which we plan to address in future work.

---

> ### Author Response · Authors · 2024-08-07
> **Rebuttal**
>
> ```
> Q8. The rationale for attention in channel space
> ```
> Modeling physical phenomena has two sets of variables, viz., spatial and temporal locations and physical variables such as temperature, pressure, etc. **Having attention in the channel enables us to model the relationships between physical variables that interact together**. Given the well-established nature of the Transformer architecture and its property to handle variable length input, it has been a natural choice for our approach. We have redesigned key components of modern transformer layers, including positional encoding, self-attention, and normalization layers, to develop codomain attention in function space.
>
> In many applications, physics problems are modeled using coupled PDEs, and generating data is often costly. Therefore, we need to have foundation models capable of tackling issues of a similar nature but with a different number of variables or changes in the variables. **Attention in the channel space helps the model to pre-train simultaneously over a wide range of PDEs with different variables without changing the architecture**. For example, fluid in the subsurface is often described by 12-14 variables that obey laws of thermodynamics, fluid dynamics, Darcy, and chemical interaction. Therefore, learning about similar physics, each with different variable counts at corresponding order, representing the particular physics, would help to train for the final subsurface model. (Please See Problem Statement (Sec 4) and lines 167-177).
>
> There are also other applications for Codano where there are different sets of physical variables with varying counts across datasets.  For example, in climate modeling, there are multiple datasets developed by different countries (please refer to CMIP6 global collaboration for climate data), each using slightly different PDEs with different variable counts. Learning the relationships among such variables represented as different channels is enabled by attention.
>
> Moreover, a recent work [1] has shown the **universal approximation property** for attention-based operators in function space, i.e., that CoDA-NO can approximate any continuous operator. Although it is not straightforward to compare different neural operators on the basis of expressivity in a general sense, our empirical studies suggest that it generalizes and performs well on a variety of tasks without any significant change to the hyper-parameters.
>
> [1] Continuum Attention for Neural Operators
> ```
> Q9. Reason for large parameterization of FNO
> ```
> We already addressed this in detail in a previous comment. For Table 2, the models are pre-trained and subsequently fine-tuned on the large PDEbench dataset. As FNO has a relatively simple architecture, we utilize a parameter-rich architecture to benefit from the pretraining and enable it to learn rich representations for subsequent prediction tasks.

---

> > ### Comment · Reviewer_ZkA5 · 2024-08-08
> >
> > Thank you for conducting the additional experiments and providing further information. I really appreciate it.
> >
> > However, I remain unconvinced that CoDA-NO is suitable for a broader range of multiphysics problems. Even in the new experiment, the FNO model trained from scratch outperforms the finetuned model with just 10 samples, which is a remarkably small number. Again, your conducted experiments suggests to me that pretraining CoDA-NO may not provide significant advantages in many cases.
> >
> > If a model trained from scratch with only 10 samples can outperform CoDA-NO, it raises concerns about the complexity of the problems being considered (since models trained from scratch with such few samples are generally not expected to be accurate), as well as the ability of CoDA-No to generalize to unseen problems.

---

> > > ### Author Response · Authors · 2024-08-10
> > > **Reply to the Reviewer's Comment-1**
> > >
> > > We thank the reviewer for the response.
> > > ## On the performance of the newly reported  Rayleigh-Bénard (RB) convection system
> > >
> > > We believe there is a misunderstanding in interpreting the results of the RB system: with 10 samples, pre-trained CoDA-NO is better than FNO and not worse.  Overall, **pretrained CoDA-NO outperforms FNO by an average of 20% on few-shot learning tasks** in these new results.
> > >
> > > We want to emphasize that **our goal isn't just for CoDA-NO to outperform FNO on a fixed dataset but to show CoDA-NO's ability to adapt to multi-physics scenarios**. Unlike FNO, which is limited to a fixed set of physics, CoDA-NO can extend from single physics (such as fluid flow) to multi-physics (fluid flow + temperature for RB system) by easily only adding new channels for additional physical variables, without any change in the backbone architecture.
> > >
> > > However, as we have already mentioned, this is a preliminary study. In particular, we could not yet simulate and train on sufficiently large datasets and tune our hyperparameters appropriately during the short discussion period. Please let us explain our process.
> > >
> > > For the new experiments, we pre-trained CoDA-NO on a compressible Navier-Stokes (NS) dataset using only one configuration with a Mach number of 0.1, $\eta = 0.01$, and $\zeta = 0.01$ (not to be confused with the NS data use in the fluid-solid interaction reported in the Table 1 of the paper). Additionally, we conducted supervised training on a Rayleigh-Bénard convection (RB) system with a Rayleigh number of 2500, applying an initial perturbation to the temperature field to initiate convection. We pre-train with **only 6000 examples with a reduced resolution of $64 \times 64$** due to time constraints.
> > >
> > > To our knowledge, there is no well-established public dataset for Rayleigh-Bernard convection systems; in particular, established benchmarks, such as PDEbench, do not include this system.
> > >
> > > It required us to generate data ourselves, and with the limited time available, we faced challenges in simulating the Rayleigh-Bénard convection system due to its computational complexity. This process requires careful choices of parameters like **temperature difference, thermal expansion, fluid height, viscosity, geometry, and initial temperature perturbations**. While we are able to generate an initial dataset, we are currently still generating RB and NS datasets with different parameters to obtain a better understanding of the pre-training capabilities of CODA-NO.
> > >
> > > In summary, the rebuttal period was too short for thorough data generation and pre-training with a large model, even though we worked continuously. Our initial results, shown in the previous response, had higher errors compared to those reported in the paper on fluid-solid interaction systems. This is why we called these results preliminary. The reason we reported those numbers was to inform the reviewers that we are dedicated to adding this study to the paper, as it will strengthen the message of this work. However, due to many challenges on the solvers' end and training overhead, we are worried this study will not be done in the next few days.

---

> > > > ### Author Response · Authors · 2024-08-10
> > > > **Reply to the Reviewer's Comment-2**
> > > >
> > > > ## Applicability to Multi-Physics Problems
> > > > In this paper, we demonstrate the applicability of the proposed architecture to a wide range of problems. In the previous response, the reviewer doubted the result from pre-training, noting
> > > > `“For the NS dataset with Reynolds number Re=400, the model trained from scratch with only 25 samples matches the performance of the pretrained model.”`
> > > >
> > > >  -- We assume the reviewer’s comment is referring to the numbers in the third column of Table 1. In this setting, the performance of CoDA-NO is 0.008 without pretraining and 0.006 with pretraining, which constitutes an improvement of 25%. So the reviewer’s comment is not true.
> > > >
> > > > Moreover, we would like to point out that, in all 9 considered setting in Table 1, we see considerable improvements of the “pretrained” models over those trained “from scratch” as: **86%, 21%, 25%, 94%, 20%, 25%, 5%, 46%, and 1%.**
> > > >
> > > > Please note that these numbers constitute **7.5x, 1.3x, 1.3x, 16.8x, 1.2x, 1.3x, 1.1x, 1.8x, 1.0x,** improvements,; which— considering the $L_2$ metric on functional data—are significant improvements.
> > > > Therefore, we respectfully disagree with the reviewer that pretraining does not provide worthwhile improvement.
> > > >
> > > > The reviewer also noted
> > > > `“In the case of NS+EW benchmark, when the Reynolds number increases to 4000, even with just 5 samples, both the finetuned and scratch-trained models exhibit similar testing errors. This suggests that pretraining may not provide significant advantages in many cases.”`
> > > >
> > > >
> > > > For this specific case, the improvement numbers for Re=4000 are **1.1x, 1.8x, and 1.0x** improvement compared to training from scratch. These numbers show that, for the case of Re=4000, **as it is out of the distribution of the pre-training data**, the gain from pretraining is not as significant as in other cases, but the model still performs better.
> > > >
> > > > Additionally, we tested our model on the well-established **PDEbench** dataset and demonstrated a **substantial 43% improvement over FNO**, a strong and well-established model, on systems like the Shallow Water Equation and Diffusion-Reaction (Table 2). This clearly shows that our method is highly effective.
> > > >
> > > > In summary, our model has proven its **superiority in tackling complex fluid-solid interactions with intricate geometries and diverse physical parameters**, delivering an **impressive 36% average improvement** (Table 1). We also demonstrate a significant improvement on the **well-established PDEbench dataset, achieving a 43% gain** (Table 2). Even with limited time and without proper pre-training, our preliminary results on the **Rayleigh-Bénard system** still showed a **20% improvement**. These results leave **no room for doubt—CoDA-NO is a powerful, versatile solution for a wide range of problems in scientific computing.**
> > > >
> > > > To this end, the paper's current results underscore the potential of our approach, not just as an incremental improvement but as a leap forward in the development of neural operator architectures. We are confident that CoDA-NO will open new avenues for research and application, particularly in complex multi-physics simulations and scientific computing.
> > > > We sincerely thank you for your time and thoughtful feedback. We hope that our response highlights the importance and impact of our contributions, and we kindly request that you reconsider the evaluation of our paper in light of these points

---

> > > > > ### Comment · Reviewer_ZkA5 · 2024-08-11
> > > > >
> > > > > Overall, I am satisfied with the rebuttal and the authors' responses. I am increasing my score to 5. Good luck!

---

### Official Review · Reviewer_qRRk · 2024-07-06

**Soundness:** 2
**Presentation:** 3
**Contribution:** 3
**Rating:** 6
**Confidence:** 4

**Summary:**

This paper presents a new operator learning method for solving multiphysics PDEs. The attention scheme is designed on channel space to capture multiple physical variables, which is called co-domain. Moreover, positional encoding and normalization layers are considered. Such a strategy enables self-supervised pretraining of PDE systems. The experiments have shown the effectiveness of the proposed method.

**Strengths:**

- The proposed idea is interesting. It enables the generalization to coupled physical systems, which is of interest to the scientific machine learning (SciML) community. Also, self-supervised pretraining is one emerging tool in SciML and will gain a lot of attention in the future.

- This paper provides the experiments on a Navier-Stokes equation and its coupled version with the elastic wave equation.

- This paper is well-organized and well-written. The details are easy to follow.

**Weaknesses:**

- This paper only considers one coupled system, i.e., NS and NS+EW. It may not validate the general applicability of the proposed method. The motivation of using this case should be enhanced. Also, considering some other PDE systems might strengthen the paper, such as the Rayleigh-Benard convection system. It is also a coupled system with NS + temperature.

- The motivation for the combination of positional encoding, self-attention, and normalization layers seems to be better clarified. Although those parts are modular (claimed in Line 68), the connections between each other are also important.

- In Appendix B.1, it would be good to include more details of self-supervised pretraining, such as masked ratio.
The evaluation metrics might not be sufficient. This paper only considers L2 errors. There are many papers considering relative l2 error [1]. For the turbulence data, researchers also care about the infinity norm a lot. It would be better to add more evaluation metrics in this paper.

**References:**

[1] Hao, Zhongkai, et al. "Dpot: Auto-regressive denoising operator transformer for large-scale pde pre-training." arXiv preprint arXiv:2403.03542 (2024).

[2] Ren, Pu, et al. "Superbench: A super-resolution benchmark dataset for scientific machine learning." arXiv preprint arXiv:2306.14070 (2023).

**Questions:**

- This paper considers the generalization to different Reynolds numbers. Is it possible to generalize to different physical parameters of elastic wave equations, such as the object size or the solid density \rho^s?

- Lines 100-101, this paper claims that it considers diverse physical systems in terms of input functions, geometries, and Reynolds numbers. I would say it’s just different PDE scenarios within one PDE type (NS and NS+EW). It seems unrelated to the diversity of PDE systems, such as reaction-diffusion, convection systems, etc.

**Limitations:**

Please see my concerns in **Weaknesses** and **Questions**.

---

> ### Author Rebuttal · Authors · 2024-08-07
>
> We appreciate that the reviewer values our work and recognizes the fact that this work is of interest to the scientific machine learning (SciML) community, presents a strategy that enables self-supervised pre-training of PDE systems, and states the importance of our experiments that have shown the effectiveness of the proposed method.
>
> ```
> Q1. Additional Coupled PDE
> ```
> We discuss the difficulty of modeling the fluid-solid interaction problem and justify our problem design in Appendix B1. Given the complex geometry and turbulent flow, the problem can be considered a challenging benchmark problem.
>
> Additionally, we provide experiments on diverse PDEs (Navier–Stokes equations, Elastic wave equations, Shallow water equations, and Diffusion reaction; please see Table 2 and Appendix Sec E, Tables 11-12) and demonstrate the architecture's performance and flexibility.
>
> As suggested, we also provide results on the Rayleigh-Benard convection system. The CoDA-NO is pre-trained in a supervised fashion on the Navier–Stokes (NS) equations. Then it is subsequently fine-tuned on a limited few shot data ($N \in \{5,10,25\}$) of the Rayleigh-Benard convection system. We show the result in the following table, where we report the $L2$ loss. We see that, in the low data regime, pre-trained CoDA-NO performs better than FNO and CoDA-NO trained from scratch. Please note that these results are primitive and yet to be finalized. Per the reviewer’s suggestion, we are running a comprehensive study on the Rayleigh-Benard convection system.
>
> | Model    | Pre-training dataset | N=5      | N=10    | N=25    |
> | -------- | --------                           | -------- |-----        |-----        |
> | FNO      | -                                | 0.439    |0.249    |0.130    |
> | CoDA-NO  | -                           | 0.315    |0.272    |0.203    |
> | CoDA-NO  | NS                      | 0.2537   |0.2223   |0.179    |
>
>
> ```
> Q2.  Motivation for the components of the CoDA-NO (The motivation for the combination of positional encoding, self-attention, and normalization layers seems to be better clarified. Although those parts are modular (claimed in Line 68), the connections between each other are also important.)
> ```
>
> We outline our motivation in the problem statement (Section 4) and explain the need for Neural Operators that can handle PDEs with varying numbers of variables (lines 167-177). Given the well-established nature of the Transformer architecture and its property to handle variable length input, it has been a natural choice for our approach. We have redesigned key components of modern transformer layers, including positional encoding, self-attention, and normalization layers, to develop codomain attention in function space. We will further emphasize this in the main text.
>
> ```
> Q3. Details of self-supervised pre-training, such as masked ratio
> ```
> We provide the masking ratio and some additional details in Appendix F. For every training sample, one of the following two masking choices is selected with equal probability
> 1.  50\% of the mesh points of 60\% of variables are masked.
> 2.  30\% of the variables are masked out completely.
>
>
> ```
> Q4. Additional Evaluation Metrics
> ```
> L2 is the generalization of RMSE to function spaces and is considered a natural metric of evaluation. In this work, we also provide an Energy Spectrum representing the correct statistical distributions of the fluids to go beyond distance-based evaluation metrics (please see Appendix D.4). L2 norms, together with the Energy Spectrum, provide considerable evidence establishing the proposed model's superiority to the baseline. In addition to these two metrics, we also include L1 and relative L2 in our list of metrics. To this end, as the reviewer knows, any evaluation metric comes with its advantages as well as limitations. We hope that the current list plays a sufficient role in establishing evident results on the superior performance of the proposed method.
>
> ### Table: L1 and relative L2 Error
>
> Results on the fluid-solid interaction dataset combining Navier-Stokes and Elastic wave equation **(NS-EW dataset)**.
>
> | Models | Pre-training Dataset | # Train = 5 (L1/Rel-L2)| # Train=25 (L1/Rel-L2)| # Train=100 (L1/Rel-L2)|
> |--------|----------------------|------------------------|-----------------------|------------------------|
> | GINO   |                      | 0.185/0.296          | 0.151/0.221          | 0.160/0.219           |
> | DeepO  |                      | 0.453/0.687           | 0.266/0.431         | 0.184/0.325          |
> | GNN    |                      | 0.083/0.130        | 0.056/0.082       | 0.059/0.082        |
> | ViT    |                      | 0.202/0.366          | 0.156/0.276         | 0.076/0.124         |
> | U-net  |                      | 0.793/1.186           | 0.284/0.463         | 0.174/0.291           |
> | Ours   |                      | 0.092/0.164         | 0.046/0.092        | 0.032/0.058          |
> | Ours   | NS                   | 0.074/0.141         | 0.032/0.072       | 0.030/0.059        |
> | Ours   | NS-EW                | 0.066/0.128         | 0.040/0.077       | 0.033/0.057         |

---

> ### Author Response · Authors · 2024-08-07
> **Rebuttal**
>
> Results on fluid motion dataset governed by Navier-Stokes equation **(NS Dataset)**.
>
> | Models | Pre-training Dataset | # Train = 5 (L1/Rel-L2)| # Train=25 (L1/Rel-L2)| # Train=100 (L1/Rel-L2)|
> |--------|----------------------|------------------------|-----------------------|------------------------|
> | GINO   |                      | 0.236/0.365          | 0.133/0.199         | 0.106/0.155          |
> | DeepO  |                      | 0.441/0.695            | 0.395/0.561         | 0.235/0.337          |
> | GNN    |                      | 0.141/0.187          | 0.074/0.096       | 0.049/0.071        |
> | ViT    |                      | 0.279/0.431          | 0.158/0.238         | 0.137/0.188           |
> | U-net  |                      | 2.001/ 3.508           | 0.683/1.178          | 0.298/0.422          |
> | Ours   |                      | 0.246/0.355          | 0.083/0.141        | 0.033/0.074        |
> | Ours   | NS                   | 0.080/0.148         | 0.040/0.081       | 0.024/0.0607        |
> | Ours   | NS-EW                | 0.075/0.143         | 0.041/0.069          | 0.022/0.057         |
>
>
> ```
> Q5. Generalization of different physical parameters
> ```
> We also consider generalization to the Reynolds number and to the inlet boundary conditions (lines 291-292). The Reynolds number is the most important factor characterizing the dynamic of the PDE, and this is reflected in different benchmarking datasets, which often contain simulations at different Reynolds numbers. Hence we followed that.
> It is possible to generalize different solid density $\rho^s$ and object sizes. However, to get a good result in the few-shot learning experiments, the pretraining data should contain PDEs with different $\rho^s$ and object sizes. We agree that this problem is a very important topic of study, deserving special treatment, and developing the right datasets for such studies is on the horizon for the ML community.
>
> ```
> Q6. Application to diverse PDE system
> ```
> We also provide experiments on different PDE systems (please see Table 2 and Appendix Section E Table 11-12), where we test the proposed CoDA-NO architecture on shallow water equation, diffusion equation, and Navier Stokes equation. Also, we agree that expanding to many more PDE systems is desirable, and for that, the field is in the process of making diverse PDE benchmark datasets.

---

> > ### Comment · Reviewer_qRRk · 2024-08-12
> >
> > Thanks for the rebuttal. The additional experiments have resolved my concerns. I will raise my score.

---

### Official Review · Reviewer_Jm1K · 2024-07-09

**Soundness:** 3
**Presentation:** 1
**Contribution:** 2
**Rating:** 6
**Confidence:** 3

**Summary:**

This paper introduces Codomain Attention Neural Operator, which tokenizes function along the channel dimension. It allows to learn representations of different PDE systems within a single model. The authors shows that finetuning a pretrained CoDA-NO on different physics yields good accuracy.

**Strengths:**

- I like the problem setting and the idea of the algorithm.
- The experimental task is quite interesting.
- The results are convincing, and the fact that the model can generalize to higher Reynold numbers seen during training is promising.
- I liked the used of GNO for handling non-uniform grids.
- The code seems solid.

**Weaknesses:**

- To me, the main weakness of the paper is that the presentation lacks of clarity. I don't see the point of doing 3 pages of mathematics in function space if, in practice, everything is done in discrete space. I think this blurs the message of the paper and it is difficult for the reader to understand what is the relevant information for understanding the actual CoDA-NO algorithm. In my opinion, these mathematics are not essential to the algorithm and could be put in appendix. I can always express a neural network architecture in function space, but since in practice we are working on discretized space, it is never done in experimental deep learning papers. Moreover, no discussion on how to go from infinite-dimensional space to discretized space is given by the authors.
This space could be used to have the actual detailed architecture. I may have missed the point on the usefulness of these sections and am willing to understand the point of view of the authors regarding this.
- I don't fully understand the CoDA-NO algorithm and I think a Figure showing the whole architecture would have clarified this.

**Questions:**

- Why do we need a VSPE per physical variable? Positional encoding are usually used when there is some sort of order between the tokens?

**Limitations:**

Overall, I think my main obstacle to provide a better score is the fact that the paper is not very clear due to the introduction of a lot of mathematics not needed to understand the algorithm in practice. These mathematics do not bring any theoretical insight of the algorithm. They are just expressing the architecture in function space. I think this space could have been used to be clearer to explain what is the architecture or the philosophy of the work. Usually, in papers where I see such mathematics, a study of the sample complexity is provided (I know it is almost impossible to do for neural networks).

I am willing to discuss these points with the authors and to modify my score accordingly.

---

> ### Author Rebuttal · Authors · 2024-08-07
>
> We appreciate that the reviewer values our work and recognizes that this work presented a method for learning representations of different PDE systems within a single model.
>
> ```
> Q1. Formulation of the Model in the Function Space and Clarity of the Paper
> ```
> We politely disagree with the reviewer. Indeed, in practice, we typically work with discretized versions of functional data. However, this does not imply that we only deal with a "fixed" finite-dimensional space for which neural networks are designed. In practice, the discretizations often vary from one sample to another (i.e., the location and number of evaluation points). Many operations that work well for "fixed" finite-dimensional data do not behave consistently across resolutions. For example, regular CNNs (or GNNs with nearest-neighbor graph) collapse to pointwise operations when the resolution is refined (see [1]) and is thus not well suited for data that comes at different resolutions.
>
> Such limitations of neural networks motivated the paradigm of neural operators as a generalization of neural networks to data originating from functions [2]. Designing architecture directly in the function space allows us to work across different discretizations seamlessly. It also allows us to draw parallels between various approaches to operator learning and ensure specific properties of the learning algorithms, such as the universal approximation property demonstrated in [3] for transformer architectures in function spaces. The mathematical foundation is often helpful in conveying the core ideas of the architecture design in operator learning. For example, here is reviewer ZkA5’s statement about our mathematical foundation: “The mathematical formulation of the proposed model is well-articulated in the paper. This clarity helps readers understand the underlying principles of the model's operation.”
>
> To better explain the architecture, following the suggestion, we provide a detailed figure of the whole architecture (see Figure 2 in the rebuttal pdf).
>
> [1] Neural Operators with Localized Integral and Differential Kernels
>
> [2] Neural Operator: Learning Maps Between Function Spaces With Applications to PDEs
>
> [3] Continuum Attention for Neural Operators.
>
> ```
> Q2. From infinite-dimensional space to discretized space
> ```
> Our implementations are based on previous works on Neural Operators [1,2,3]. These studies discuss how operations on functional spaces are approximated on discretized data, e.g., using Riemann sum approximation of the integral operator, which scales appropriately with the resolution. We will clarify this concept in the paper.
>
> [1] Fourier Neural Operator for Parametric Partial Differential Equations (see Section 4).
>
> [2] Neural Operator: Graph Kernel Network for Partial Differential Equations (see Section 3).
>
> [3] Neural Operator: Learning Maps Between Function Spaces With Applications to PDEs.
>
> ```
> Q3. Figure showing the whole architecture.
> ```
> Thanks for the suggestion. We added a detailed figure (Figure 2) in the rebuttal pdf.
>
> ```
> Q4. Use of VSPE per physical variable
> ```
> In this work, we treat each of the input function's variables (codomains) as a token for the Codomain attention. The model needs a way to identify each variable (e.g., temperature, velocity, pressure). In LLM, it is done through position ID. In CoDANO, we propose the Variable-Specific Positional Encoding (VSPE), which is learned separately for each variable and then concatenated with the corresponding variables. It allows the model to differentiate between the various PDE variables and their interactions. The following table demonstrates the benefit of using VSPE for fine-tuning on $5$ examples for $\mu = 5.0$ (from Appendix D1, Table 3). Please refer to Appendix D1, Table 3 for a detailed analysis.
>
> | VSPE         | Pretrain Dataset | NS    | NS+EW |
> |--------------|------------------|-------|-------|
> | x            | NS               | 0.049 | 0.079 |
> | x            | NS EW            | 0.045 | 0.057 |
> | $\checkmark$ | NS               | 0.025 | 0.071 |
> | $\checkmark$ | NS EW            | 0.024 | 0.040 |
>
> ```
> Q5. On the clarity and organization of the paper
> ```
> We have addressed this in a previous comment, noting that developing architecture directly in the function spaces is crucial, conveying the philosophy of the work. We also agree with the reviewer that this should not overshadow the practical implementation aspect of the work. Our philosophy and goals are outlined in the introduction (lines 28-41) and in the Problem Statement (see Sec 4).
>
> We provide additional detailed figures (see the attached PDF), and we move some background details to the Appendix, as suggested by the reviewer.
>
> To this end, we would appreciate it if the reviewer reconsidered our explanation and problem setup of operator learning in this paper. The field of neural operators is still in its early stages, and similar to the early stages of neural networks, explaining the problem setup and detail of the architecture may still be considered crucial and insightful.

---

> > ### Comment · Reviewer_Jm1K · 2024-08-12
> >
> > I would like to thank the authors for their rebuttal.
> >
> > Q 1-2. Thank you for the thorough justification of why all these mathematics are in the main paper. I think I get the point, and I strongly suggest to write a high-level explanation motivating their introduction. Indeed, in such an experimental paper, one may ask themselves the same questions as mine, and be discouraged of reading it if it is not motivated enough.
> >
> > Q3. Thank you for the figure. I think this is a bit better now, but I really want to insist on the fact that clarity about the architecture is key in these complex experimental papers, and not making it clear would discourage people wanting to enter the field.
> >
> > Q4. Thank you for this clarification.
> >
> > Q5. I hope the reorganization by the authors could make the method easier to understand.
> >
> > Overall I think this is a good paper tackling a new problem. The paper is not very clear, but the authors seem to be willing to arrange the paper to make it more accessible. I will increase my score to 6, and discuss with the other reviewers and the AC to have their opinion.

---

> > > ### Author Response · Authors · 2024-08-13
> > >
> > > Thank you for your response and valuable suggestions. We are delighted to see that the reviewer has appreciated our work.
> > >
> > > Response to Q1-2: We will enhance the high-level explanation in the introduction.
> > >
> > > Response to Q3: In addition to adding the extended figure, we will incorporate pseudocode in the revised text to convey the procedure clearly. We believe including the extended figure and pseudocode will effectively communicate our idea.
> > >
> > > Response to Q5: As indicated in the rebuttal, we will relocate some background information to the appendix and concentrate more on the motivation and high-level procedure in the main text.

---

### Official Review · Reviewer_rAy3 · 2024-07-13

**Soundness:** 3
**Presentation:** 3
**Contribution:** 3
**Rating:** 7
**Confidence:** 3

**Summary:**

The authors propose CoDA-NO, a neural operator architecture that captures interactions across different physical variables of coupled PDE systems. The method involves a generalization of the transformer architecture, including self-attention, positional encodings, and normalization, to function spaces. On two novel datasets for fluid-structure interaction and fluid dynamics, the authors show that their method achieves state-of-the-art performance.

**Strengths:**

- The paper investigates an interesting problem of how to appropriately capture interactions across different physical variables, that allows for generalization to new codomains.
- As far as I am aware, the generalization of the Transformer architecture to function spaces is novel.
- The experimental results, especially the generalization capabilities (from fluid dynamics to fluid-solid interactions) are impressive.
- Ablation studies on the proposed architectural changes are thorough.

**Weaknesses:**

Overall, the experiments seem quite compelling. However, it could be illuminating to provide a graphical visualization of the data from Table 1, regarding efficiency of fine-tuning and robustness to out-of-distribution inputs: see questions.

**Questions:**

- It seems that the performance of the models across the board continue to improve with increase few-shot fine-tuning samples beyond N=100. What does the scaling look like for the proposed model and where does performance saturate?
- Similarly, the model is evaluated on the in-distribution Re=400 and the out-of-distribution Re=4000 settings, for which the performance of the model is comparable. What does the scaling look like as the task becomes further out-of-distribution (e.g. decreasing velocity)?

**Limitations:**

The authors address the limitations.

---

> ### Author Rebuttal · Authors · 2024-08-07
>
> We appreciate that the reviewer values our work and recognizes the main contributions to the novel generalization of the transformer architecture, including self-attention, positional encodings, and normalization to function space, along with the introduction of two new challenging datasets. We further may state that in this work, we not only based our study on the mentioned two PDEs, but we also have SWE and Diffusion Eq. Per the reviewer's request, we are running experiments on the Rayleigh-Benard convection system, with initial results stated in the reviewer qRRk’s reply.
> ```
> Q1 Visualization of the Data
> ```
> We thank the reviewer for the visualization suggestion. We provide the visualization of data in the Appendix, Sec B Dataset Description.
>
> ```
> Q2 Performance scaling with data (It seems that the performance of the models across the board continues to improve with an increase in few-shot fine-tuning samples beyond N=100. What does the scaling look like for the proposed model, and where does performance saturate?).
> ```
> We conduct experiments with N=250, 500, 1000 for NS+EW with Re=400. We observe that performance continues to improve when training on more data. Therefore, finding the saturation point requires the generation of a much larger dataset, which requires future development and compute allocation for the solver. The precise scaling is problem-dependent.
>
> | Model    | Pre-training    | N = 100  | N=250    | N=500    | N=1000    |
> | -------- | --------        | -------- |----------|----------|-----------|
> | CoDA-NO  | NS              | 0.005    | 0.003    | 0.003    | 0.001     |
> | CoDA-NO  | NS+EW           | 0.003    | 0.003    | 0.002    | 0.001     |
>
> ```
> Q3 Similarly, the model is evaluated on the in-distribution Re=400 and the out-of-distribution Re=4000 settings, for which the performance of the model is comparable. What does the scaling look like as the task becomes further out-of-distribution (e.g. decreasing velocity)?
> ```
> For Re=4000, we observe that the performance of the pre-trained models is worse than for Re=400. We hypothesize that the reason may be intuitively twofold,
> 1.     Fluid motion at Re=4000 is highly turbulent and
> 2.     the data is out-of-distribution.
> Inspired by other studies in machine learning, CoDA-NO will perform worse on out-of-distribution data and will require further data for finetuning. However, our evidence suggests that the performance will be better than training from scratch.
>
> Also, finding the accurate scaling with respect to ODD, i.e., what degree of out-of-distribution the model can handle, requires extensive problem-dependent study

---

> > ### Comment · Reviewer_rAy3 · 2024-08-12
> >
> > Thanks to the authors for the detailed response and additional experimental results! I will leave my score as is.

---

### Author Rebuttal · Authors · 2024-08-07

We thank the reviewers for their valuable comments. We appreciate reviewers for recognizing the presentation of a novel neural operator architecture that “captures interactions across different physical variables of coupled PDE systems and the fact that the method involves a generalization of the transformer architecture, including self-attention, positional encodings, and normalization, to function spaces. “ It has been recognized by reviewers that this study is of interest to the scientific machine learning (SciML) community.

This paper also introduces two challenging datasets based on NS and NS+EW, which are coupled PDEs that resemble problems in weather forecasting and climate modeling.

Based on the reviewer's comments, we have the impression that reviewers may have interpreted that we only study CoDA-NO on the two mentioned datasets. We would like to state that we study CoDA-NO on four PDEs: NS, NS+EW, SWE, and Diffusion equations. Accordingly, we edited the paper's presentation, ensuring that the explanations and structure imply that we study more than two PDEs.

In addition, per the reviewer qRRk’s suggestion, we started to incorporate another PDE system, i.e., the Rayleigh-Benard convection system. Our first runs show positive results, as presented below, showing an **average improvement of 20% compared** to the Fourier neural operator (see the response to reviewer qRRk). We will add the finalized comprehensive study to the main draft.

---

### Decision · Program_Chairs · 2024-09-25

**Decision:**

Accept (poster)

**Comment:**

The paper introduces a neural operator model for solving systems of coupled differential equations, capturing the interaction between the physical variables. It leverages several novel extensions of classical transformer components. The authors evaluate their model on representative datasets and also introduce two novel datasets developed specifically for the paper.

The reviewers agree on the importance of the problem, the novelty of the proposed model, and the significance of the experiments. During the rebuttal, the authors presented complementary experiments, including one on a new dataset of coupled equations. Three out of four reviewers increased their scores after the rebuttal discussion. The authors are encouraged to follow the reviewers' suggestions to improve the clarity of the presentation.